# Effect of Temporal Sampling Interval on the Irradiance for Moon-Based Wide Field-of-View Radiometer

**DOI:** 10.3390/s22041581

**Published:** 2022-02-17

**Authors:** Yuan Zhang, Shengshan Bi, Jiangtao Wu

**Affiliations:** Key Laboratory of Thermo-Fluid Science and Engineering, Ministry of Education, School of Energy and Power Engineering, Xi’an Jiaotong University, Xi’an 710049, China; yuanzhang@stu.xjtu.edu.cn (Y.Z.); jtwu@mail.xjtu.edu.cn (J.W.)

**Keywords:** irradiance, radiation budget, Earth observation, Moon-based, sampling interval, CERES

## Abstract

Moon-based Earth radiation observation can provide longer-term, continuous multi-angle measurements for the Earth’s outward radiative flux. In addition, the large distance between the Moon and Earth means that the radiation can be monitored by a non-scanning Moon-based Wide Field-of-View (MWFOV) radiometer considering the Earth as one pixel. In order to parameterize the radiometer, studying the effect of the temporal sampling interval on irradiance is of great importance. In this work, based on radiation transfer model, simulated irradiance time series from March 2000 to December 2020 were analyzed. Then, we used them to reveal the effects of the sampling interval on irradiance. The results show that the measurements of the MWFOV radiometer can reveal the variation of irradiance on hourly, daily and monthly time scales, and the high-frequency measurements can reflect the variation of scene types in the MWFOV-viewed area. In order to obtain more meaningful measurements, the radiation resolution of the MWFOV radiometer should be better than 0.5 mW∙m^−2^ with an accuracy of 1% or better in the future actual design, and the sampling interval should be less than 1 h, which can ensure that 97% of the surface area can be sampled more than nine times per day for longwave radiation. The derived results in this study could facilitate Moon-based data processing and the determination of the sampling interval and radiation resolution of an MWFOV under a certain manufacturing cost and error limit.

## 1. Introduction

Global climate change exerts important impacts on the development of the world’s economy and society. In essence, global climate change is driven by the Earth Radiation Budget (ERB) at the Earth’s top of the atmosphere (TOA), so the long-term monitoring of the ERB is of fundamental importance for understanding climate change and analyzing the Earth’s energy imbalance [1,2]. The ERB quantifies how the Earth gains energy from the Sun (*F_0_*) and loses energy to space through outgoing shortwave (SW) radiation (OSR) and outgoing longwave (LW) radiation (OLR), so it is of fundamental importance for the climate and for predicting future global climate change. In addition, according to the conservation of energy, we can obtain the equation *F*_net_ = *F*_0_ − OSR − OLR. *F*_net_ is the net radiation on the Earth’s TOA between incoming and outgoing radiative fluxes. OLR is the energy radiating from the Earth atmosphere system in the spectral region from 5 to 200 μm, and OSR is the radiative flux reflected by the Earth atmosphere system in the spectral region from 0.2 to 5 μm. The incoming and outgoing radiation at the Earth’s TOA is usually measured from space with dedicated remote sensing instruments, such as Earth Radiation Budget Experiment (ERBE), which was used in the early days of observations [3,4,5]. As an improved version of ERBE, the Clouds and the Earth’s Radiant Energy System (CERES), onboard the Terra and Aqua satellites [6], used two complementary orbits for the sampling of the diurnal cycle to measure the TOA radiative fluxes and atmospheric radiation [6,7]. The Geostationary Earth Radiation Budget (GERB) instrument carries out the dedicated observation of the ERB from a geostationary orbit, and it helps to mitigate the lack of temporal samples in the low-orbit satellite observation data [8,9,10]. The Deep Space Climate Observatory (DSCOVR), launched by NASA in 2015, is situated at the Earth–Sun Lagrange Point (L1) and can observe the LW and SW radiation from the illuminated side of the Earth with high accuracy and sampling frequency; however, due to the limitation of the orbit, only about 92–97% of the sunlit Earth is visible to the National Institute of Standards and Technology Advanced Radiometer (NISTAR) [11]. Although satellite-based observations have enhanced our understanding of the ERB, they still have several limitations such as limited longevity, the limited instantaneous field of view of the Earth, limited temporal sampling and the challenging nature of synthesizing a global image from heterogeneous satellite datasets [12,13].

As the only natural satellite of the Earth, the nearside of the Moon would be suited to monitoring the Earth’s outgoing radiation [14,15]. Due to the limitation of synchronous rotation and the large Earth–Moon distance, the Moon can observe nearly half of the Earth, and the data inconsistency resulting from the viewing geometry difference is much reduced for Moon-based Earth radiation observation compared with satellite observations systems. The latitudes of the Moon’s nadir point on the Earth’s TOA vary between 28°43′ N and 28°43′ S, so it can also provide continuous observations of the majority of observed points with continuously changing angles [16,17]. The huge lunar surface space also provides a large number of options for various types of equipment to collect various types of data [13]. In addition, the long life cycle and stable orbit of Moon-based platforms enable long-term, time-series, high-frequency observations to be recorded, covering all local times and all seasons in a year. Of course, the observation instruments on the lunar surface also face many challenges, such as the temperature difference between day and nighttime [18], the high-energy particles from outer space and Moon dust and energy. However, similar situations also appear for satellite-based platforms, and many technologies have been invented and applied to overcome these challenges, which are still suitable as sensors on the lunar surface [19,20].

Until now, a number of studies have been conducted considering Moon-based Earth observations. The research mainly focuses on the feasibility and effectiveness of Earth observation from a Moon-based platform [16,21], model simulations based on geometric conditions [17,22] and the potential applications of Moon-based Earth observations [23,24]. Huang (2008) [25] pointed out that variation in the lunar surface temperature is mainly controlled by terrestrial radiation during the lunar nighttime, which provides evidence of feasibility for using a Moon-based platform to study the ERB. Guo et al. (2016) [26] proposed that a Moon-based platform can provide large-scale, continuously changing observation angles and long-term, high-quality data about the outgoing radiation of the Earth. Duan et al. (2019) [27] built an irradiance-estimating model for terrestrial radiation at the entrance pupil of a simplified, single-pixel, Moon-based Earth radiation observatory instrument and predicted the range and magnitude of irradiance. Ye et al. (2019) [28] conducted temporal sampling error analysis of the Earth’s outgoing radiation based on the datasets from NASA’s Goddard Earth Observing System Version 5, but the analysis was performed at the level of the top of the atmosphere. Measurements of OSR and OLR with dedicated broadband instruments started on Nimbus 6 in 1975 [25]. Two types of instruments are contained in the Nimbus 6: a non-scanning wide field-of-view (WFOV) instrument measuring the radiation of the Earth from limb to limb and a scanning narrow field-of-view (NWOF) instrument measuring the radiation with higher resolution [29]. Moreover, after 1975, various types of wide-field-of-view radiometers were launched, such as the Earth Radiation Monitor on the FengYun (FY)-3 series [30], the shortwave outgoing radiation monitor (IKOR) on the Meteor series [31], Radiometer Assessment using Vertically Aligned Nanotubes (RAVAN) onboard a 3U CubeSat [32] and the National Institute of Standards and Technology Advanced Radiometer (NISTAR) on the DSCOVR [11]. In addition, Huang et al. (2021) [12] suggested that the first-generation Moon-based Earth Radiation Budget Experiment (MERBE) instruments are similar to the latest version of the CERES instrument, and both are wide-band radiometers. Similar to the RAVAN and NISTAR, placing a dedicated remote sensing instrument on the lunar surface, the large distance between the Moon and Earth (380,000 km on average) means that outgoing radiation from the Earth’s TOA can be monitored by the non-scanning Moon-based Wide Field-of-View (MWFOV) radiometer considering the Earth as one pixel [33,34], and the distance also brings great difficulties to detection by the scanning method. The single-pixel wide field-of-view observations also mitigate the lack of temporal samples caused by orbit limitations and the error due to time interpolation [2].

Outgoing radiation is a key part of research into the ERB at the Earth’s top of the atmosphere. Irradiance is the radiation that arrives at the entrance pupil plane of the instrument, and it finally arrives at the plane of the detection element. The irradiance is converted to a digital signal; afterwards, the digital signal is processed and saved by the electrical system [2,32]. Since ERB measurement equipment is usually a heat detector, its sampling frequency is quite different, usually a few seconds [7] (the Clouds and Earth’s Radiant Energy System (CERES)) or a few minutes [8,9,10] (the Geostationary Earth Radiation Budget (GERB)) or even a few hours [11] (the NISTAR onboard the Deep Space Climate Observatory (DSCOVR)). Therefore, the ascertainment of a sampling scheme for an MWFOV radiometer is of great importance. Current LEO ERB systems cannot record the rapid variability of the OLR and OSR of the Earth system due to the limited temporal sampling coverage. The sampling interval represents the time difference that the adjacent samplings have. A shorter sampling interval would result in more samples every day, but the radiometer would need a more precise design and have a higher cost. The sampling temporal sequence at a certain sampling interval would also have consequences for the measurement error, since it has an effect on the final temporal interpolation for the utilization of data. In addition, the sampling frequency is closely related to the response time of the detection element. The smaller the response time, the higher the sampling frequency. As a larger signal-to-noise ratio (SNR) for the MWFOV radiometer will help to obtain higher-quality data and more useful information, the improvement of the SNR is of great importance. The quantitative analysis of the temporal sampling characteristics of the irradiance will help in the selection of an optimal detection element to obtain the highest SNR for an MWFOV radiometer. However, the effects of the sampling interval on the irradiance for the Moon-based Wide Field-of-View Radiometer remain poorly understood. Therefore, the effect of the temporal sampling interval on the irradiance for the Moon-based Wide Field-of-View radiometer is analyzed in this work, and the derived results could facilitate the ascertainment of a sampling scheme for an MWFOV radiometer under a certain manufacturing cost.

In this work, based on the radiation transfer model, the effect of the temporal sampling interval on the irradiance for a Moon-based Wide Field-of-View radiometer is analyzed. In Section 2, we describe the Moon-based Earth observation geometry, the calculation process of irradiance and the analytical method. Section 3 shows all of the results, and Section 4 presents a discussion. In Section 5, the concluding remarks are presented.

## 2. Materials and Methods

The goal of this work is to investigate the effect of the temporal sampling interval on irradiance for a one-pixel Moon-based Wide Field-of-View (MWFOV) radiometer. Therefore, in this work, on the basis of the entrance pupil irradiance estimating model, the simulated original irradiance time series were obtained by using the CER_SYN1deg-1Hour_Edition4 data products from March 2000 to December 2020 and SW and LW ERBE angular distribution models (ADMs). In addition, the original irradiance time series was taken as the substitute for the truth in the following analysis.

### 2.1. Moon-Based Earth Observation Geometry

In order to obtain the irradiance time series, the geometric relationship between the MWFOV radiometer and the Earth in a unified coordinate system was built. The Planetary and Lunar Ephemerides DE430 was used to acquire the position parameters. Figure 1 presents the geometrical relationship between the Earth, the Sun and the MWFOV radiometer. Due to the motion of the Moon’s orbit plane and the Earth’s rotation, the MWFOV-radiometer-viewed area on the Earth’s TOA varies with time. Furthermore, as the Moon–Earth system orbits the Sun, the Sun-illuminated part on the Earth’s TOA is also temporally variable. Usually, the instantaneous MWFOV-radiometer-viewed field includes a sunlit portion and a dark portion (as shown in Figure 1). The sunlit portion emits not only the OLR, but also the OSR reflected by the Earth’s TOA. The dark portion only has the OLR. Since the rotation period of the Earth is 24 h, the nearside of the moon can sample almost the whole Earth each day [14]. For the ERB at the top of the atmosphere, the influence of clouds on the radiative fluxes through the Earth–atmosphere system is one of the major sources of estimation uncertainty. The radiative energy flow from the Earth’s surface plays an important role in the study of land surface temperature (LST) and the ERB at the surface. So, the measurements of the thermal radiation in the narrow spectral region 8−12 μm “window” (NW) are expected to improve the estimates of LST, ERB and the clear-sky longwave flux over nonblack land surfaces [6,12]. In addition, Huang et al. (2021) [12] proposed the Moon-based Earth Radiation Budget Experiment (MERBE) and suggested that the first-generation MERBE instruments are similar to the latest version of the CERES instrument with three spectral ranges: total (0.2–200 µm), window (8–12 µm) and shortwave (0.2–5 µm). Thereby, based on these features, the analysis of irradiance in this work includes the outgoing shortwave radiation (OSR), the outgoing longwave radiation of the ”window” channel (OLR-NW) and the outgoing longwave radiation (OLR).

### 2.2. Radiation Transfer Function

The instantaneous irradiance, Φ, measured by the MWFOV radiometer at latitude *θ*_Moon_ and longitude *λ*_Moon_, is the integral of the radiation from all sites within the MWFOV-radiometer-viewed region. Moreover, the curvature of the MWFOV-radiometer-viewed area will mean that the contributions of different Earth TOA sites in the MWFOV-radiometer-viewed area to irradiance are spatial variables. Therefore, the irradiance of SW (Φ_SW_), LW (Φ_LW_) and NW (Φ_NW_) for the MWFOV radiometer can be derived from
(1)$$\Phi_{\mathrm{SW}}=\iint_{A}\int_{\Omega }L_{\mathrm{SW}}(\theta_{0},\theta ,\phi )dAd\Omega$$
(2)$$\Phi_{\mathrm{LW}}=\iint_{A}\int_{\Omega }L_{\mathrm{LW}}(\gamma ,\theta ,t)dAd\Omega$$
(3)$$\Phi_{\mathrm{NW}}=\iint_{A}\int_{\Omega }L_{\mathrm{NW}}(\gamma ,\theta ,t)dAd\Omega$$
where *L*_SW_ (*θ*_0_, *θ*, *ϕ*) is the shortwave radiance at solar zenith angle *θ*_0_, viewing zenith angle *θ* and relative azimuth angle *ϕ* (see Figure 2a). *L*_LW_ (*γ*, *θ*, *t*) and *L*_NW_ (*γ*, *θ*, *t*) are the longwave radiance and NW radiance, respectively, at colatitude *γ*, viewing zenith angle *θ* and time *t* (see Figure 2b).

To derive the irradiance, the temporal–spatial distribution of the MWFOV-radiometer-viewed area and the sunlit portion at the Earth’s TOA should be first derived. Due to the difficulties in solving actual Equations (1)−(3), we discretized the Earth’s TOA into various elements, and the numerical integration method was utilized. The observation geometry for the calculation of the irradiance is presented in Figure 2. The rationale of the numerical integration method (see Equations (4) and (5)) is to sum the individual contributions of every discrete grid node in the MWFOV-radiometer-viewed area to the instrument’s irradiance (see Figure 2). Since the Earth−atmosphere system is not a black body, the reflected shortwave radiation by the Earth’s TOA not only depends on the incident spectrum and the incident direction but also on the features of the surface and viewing angle. However, the outgoing longwave radiation depends on time and the features of the surface and viewing angle. In addition, identical to the similarity between the “window” and the longwave channel, the same expression is used (see Equation (5)), and the OLR-NW is used to represent the irradiance of the ”window” channel for convenience in the following analysis. Based on satellite platform data such as CERES, the individual contribution of the Earth’s TOA to irradiance can be derived from the TOA radiation flux and the angular distribution models (ADMs). For convenience, the size of the discrete grid is set to be equal to that of the CERES synthetic datasets. Therefore, the irradiance of SW (Φ_SW_) and LW or NW (Φ_LW/NW_) can be derived from the following discrete summation equations [35,36]:(4)$$\Phi_{\mathrm{SW}}=\sum_{N}\frac{L(\theta_{0,i},\theta_{i},\phi_{i})}{D_{i}^{2}}dA_{i}\cos\theta_{i}\cos\beta_{i}$$
(5)$$\Phi_{\mathrm{LW/NW}}=\sum_{N}\frac{L(\gamma_{i},\theta_{i},t_{i})}{D_{i}^{2}}dA_{i}\cos\theta_{i}\cos\beta_{i}$$
where *D_i_* is the distance between the site of the observation instrument and the discrete grid node *i*. *β_i_* is the viewing zenith angle, as shown in Figure 2.

### 2.3. ADMs and Datasets

For an imaginary surface element at the top of the atmosphere, the ADMs are defined as the ratio of the radiance for each direction out to space and the total hemispheric flux leaving the surface element [3,37,38,39,40]. Additionally, it is necessary to classify the Earth observations into a set of scenes for the successful application of the ADMs by taking the variation of scene in the field of view into account. Information about the angular characteristics of the reflected shortwave radiation and emitted longwave radiation from the Earth–atmosphere system [40] is required for the calculation of irradiance. The angular directional models used for ERBE are derived from the Nimbus 7 Scanner instrument and Geostationary Operational environmental satellite (GEOS) instruments [3,41,42]. The maximum likelihood estimation method, surface features and cloud features are used for the scene identification of ERBE ADMs, and the Earth observations are divided into 12 scenes, as shown in Table 1 [41,42]. The ADMs for outgoing shortwave (SW) and longwave (LW) radiation are defined according to the angular coordinate system shown in Figure 2. The LW and SW anisotropic function *R*_SW_ and *R*_LW/NW_ can be expressed as
(6)$$R_{\mathrm{SW}}(\theta_{0},\theta ,\phi )=\frac{\pi L(\theta_{0},\theta ,\phi )}{M(\theta_{0})}$$
(7)$$R_{\mathrm{LW/NW}}(\gamma ,\theta ,t)=\frac{\pi L(\gamma ,\theta ,t)}{M(\gamma ,t)}$$
where *M* (*θ*_0_) is the SW-equivalent Lambertian flux at solar zenith angle *θ*_0,_ and *M* (*γ*, *t*) is the LW-equivalent Lambertian flux at colatitude *γ* and the time *t*. In addition, LW ERBE ADMs are also used for the calculation of irradiance for the NW channel for convenience. The ray from the Sun to the target grid area and the normal distribution of the target surface form the principal plane, and the relative azimuth angle *ϕ* is the angle between the principal plane and exiting ray, as shown in Figure 2a. In contrast to *R*_SW_, *R*_LW_ considers time variation and divides the time into four seasons (according to the Northern Hemisphere): spring (March, April and May), summer (June, July and August), autumn (September, October and November) and winter (December, January and February).

In this study, the ERBE ADMs are utilized to simplify and complete the calculation. The anisotropic function *R*_SW_ for the scenes of clear ocean and clear land are shown in Figure 3a,b. The results show that the anisotropic factors (AFs) rise with the increase in the sensor’s viewing zenith angle. The geometric relationship of the Sun, the target viewed area on the Earth’s TOA and the sensor’s position on the lunar surface have a strong influence on the AFs. The decrease in the viewing zenith angle and the increase in the relative azimuth angle for the MWFOV radiometer will cause the SW AFs to increase. The statistical results of SW AFs show a variation range of 0.41~12.76 for clear ocean and 0.666~5.62 for clear land, respectively. The smaller the solar zenith angle, the smaller the variation range of AFs; for example, the range is 1.081~2.151 for clear ocean and 1.084~1.17 for clear land when the solar zenith angle ranges from 0° to 25.84°. Compared with the *R*_SW_, the LW AFs’ *R*_LW_ has a small variation range, and *R*_LW_ will decrease with the increase in the sensor’s viewing zenith angle. Although the Earth–atmosphere system is complex and constantly changing, the variation in time has less influence on the *R*_LW__,_ and the trend is consistent in different seasons. The *R*_LW_ ranges from 0.85 to 1.1 under different scene conditions and times (as shown in Figure 3c,d).

The dataset of SYN1deg-1hourly Ed4A (provided by the National Aeronautics and Space Administration Langley Research Center’s (LaRC) atmospheric science center and obtained from https://asdc.larc.nasa.gov/project/CERES) is used to obtain the radiation flux data at the node i. This dataset was collected using several instruments on multiple platforms, such as CERES Imaging Radiometers on Geostationary Satellites and the Moderate-Resolution Imaging Spectroradiometer (MODIS) on Terra and Aqua [43,44]. Data collection for this product is ongoing. The CERES missions are a follow-on to the successful Earth Radiation Budget Experiment (ERBE) mission and provide a relatively accurate measurement of multiple Earth radiation parameters including the TOA outward radiation flux and cloud properties. Therefore, the CERES has the potential to help us to better understand the ERB [45]. The CERES Synoptic 1 degree (SYN1deg) products could provide the highest temporal resolution for the TOA flux dataset with a 1° latitude × 1° longitude spatial resolution. The CERES SYN1deg-1Hour Ed4A products are designed to provide the highest temporal resolution TOA flux dataset by incorporating hourly GEO imager data and by taking advantage of the additional GEO imager channels to improve the cloud property retrievals, computed surface fluxes and GEO-derived TOA fluxes.Therefore, to investigate the effect of the temporal sampling interval on the irradiance for the one-pixel Moon-based Wide Field-of-View (MWFOV) radiometer, the SYN1deg-1hourly Ed4A product is used in this work.

### 2.4. The Simulated Original Irradiance Time Series

To obtain the simulated original irradiance time series by using radiative flux from CERES data products, the numerical integration method is used, and the MWFOV-radiometer-viewed area at the Earth’s TOA is subdivided into finite geographic elements. Then, the original irradiance time series can be obtained by summing up the individual contributions of these elements. The individual contributions from every finite geographic element can be calculated by utilizing the radiation transfer function after considering the radiance, cosine weighting, solid angle, TOA radiation flux and ADMs. Here, the ADMs are used to complete the conversion process between the measured TOA radiances and the TOA radiation flux data. So, discrete radiation transfer functions (4) and (5) can be further expressed as the following Equations (8) and (9):(8)$$\Phi_{\mathrm{SW}}=\sum_{N}\frac{M(\theta_{0,i})R_{\mathrm{SW}}(\theta_{0,i},\theta_{i},\phi_{i})}{\pi D_{i}^{2}}dA_{i}\cos\alpha_{i}\cos\beta_{i}$$
(9)$$\Phi_{\mathrm{LW/NW}}=\sum_{N}\frac{M(\gamma_{i},t_{i})R(\gamma_{i},\theta_{i},t_{i})}{\pi D_{i}^{2}}dA_{i}\cos\alpha_{i}\cos\beta_{i}$$
where *M* (*θ_0, i_*) and *M* (*γ_i_*, *t_i_*) in Equations (5) and (6) are the SW and LW equivalent Lambertian flux of discrete node *i* at the Earth’s TOA.

The flowchart for the calculation of the simulated original irradiance time series of the MWFOV radiometer is shown in Figure 4. The first step is to set the time range, time step and lunar surface position. The second step is to discretize the Earth’s TOA at a resolution of 1° × 1° and acquire the coordinate vector in the same coordinate and time system based on the Moon-based Earth observation geometry. The third step determines the MWFOV-radiometer-viewed node area at the Earth’s TOA. The fourth step solves the radiation transfer function to obtain the MWFOV radiometer’s entrance pupil irradiance at a certain moment. The fifth step judges whether the calculation is completed at all time points. If yes, the irradiance is output; if no, then the next time calculation is entered—that is, the model returns to the second step. Finally, the simulated irradiance time series is output, and the calculation process is ended. Due to the maximum difference between different lunar locations being 9 × 10^−4^ W∙m^−2^, which is too small to remarkably improve the observation performance of the platform, the lunar surface site 0° E 0° N is selected as the position of the MWFOV radiometer for the study of the effect of the temporal sampling interval on irradiance.

### 2.5. Methodology

#### 2.5.1. Temporal Sampling Method and Metrics

The theoretical simulation model for observing the Earth’s outgoing radiation from a Moon-based platform is introduced in Section 2.1, Section 2.2, Section 2.3 and Section 2.4. In this section, the methodology is carried out to study the irradiance by evaluating the temporal sampling errors. The simulated original irradiance time series with a sampling interval of 1 h is called the original time series. The original time series provides a suitable substitute of the truth measured by an MWFOV radiometer, and the subsampling of the original time series with a specific frequency mimics the observations of the radiometer. The effect of the temporal sampling interval on the irradiance is to quantify the errors caused by the subsampling of the original time series.

As the temporal resolution of CERES SYN1deg Ed4A is one hour, the simulated irradiance is output every 60 min. Let Φ be the simulated irradiance time series of the outgoing radiation (SW, LW or NW) from the Earth–atmosphere system; then,
(10)$$\Phi_{b,t_{i}}=\{\Phi_{b,t_{1}},\Phi_{b,t_{2}},\ldots ,\Phi_{b,t_{i}},\ldots ,\Phi_{b,t_{n}}\}$$
where subscripts *t_i_* (*i* = 0,1,2, 3…) and *b* denote the output times and the band.

For a particular sampling frequency, a subsampled time series can be constructed by choosing the specific starting point. For example, for a 2-h sampling interval, the subsampled time series are 0, 2, 4, …, 20, 22 and 1, 3, 5, … 21, 23. For the specific starting point, a subsample can be constructed by choosing a particular sampling frequency, such as 0, 2, 4, …, 20, 22 and 0, 3, 6, …, 21, 24. Let $\Phi_{b,t_{i}}^{n,j}$ be the subsampled time series corresponding to the *j*-th (*j* = 0, …, 2*n*) starting point, where *n* (*n* = 1, …, 24) is the sampling frequency in hours; then,
(11)$$\Phi_{b,t_{i}}^{n,j}=\{\Phi_{b,t_{1}}^{1,j},\Phi_{b,t_{2}}^{2,j},\ldots ,\Phi_{b,t_{i}}^{n,j},\ldots ,\Phi_{b,t_{1}}^{n,1},\Phi_{b,t_{2}}^{n,2},\ldots ,\Phi_{b,t_{i}}^{n,j}\}$$

For example, for the 3-th starting point and the sampling frequency of 2 h, the subsampled time series is as follows:(12)$$\Phi_{b,t_{i}}^{n,j}=\{\Phi_{b,t_{1}}^{2,3},\Phi_{b,t_{2}}^{2,3},\ldots ,\Phi_{b,t_{i}}^{2,3}\}$$

To quantify the effect of the subsampling, the metrics of the uncertainties in the mean, the absolute error of the subsamples and the correlation between the subsamples and the original time series are adopted [46,47,48]. As mentioned above, multiple time series can be constructed with the variation of the sampling interval and start time. The daily mean of each time series is computed by Equation (13), and the method is the same for monthly and annual means.
(13)$${\bar{\Phi }}_{b,t_{i}}^{n,j}=\frac{\sum_{N}\Phi_{b,t_{i}}^{n,j}}{N}$$

Different subsampled time series have different means, and the variation in these means can be described as the uncertainty in subsampling. Figure 5 provides an example of the monthly means in October 2017 for all of the realizations from the various sampling intervals and various start points. Here, the monthly means are plotted against sample frequency. When the sampling frequency is less than 4 h, the spread of monthly means is quite small. In addition, if the sampling frequency is greater than 12 h, the error of measurements will exceed 1.75 mW∙m^−2^. When the sampling frequency is 24 h, only one location is effectively sampled per day, and the error spread is large. Moreover, compared to the monthly means, the daily means will have a larger and faster spread.

For a specific interval and a start point, the standard deviation of the difference between the original and subsampled time series over all possible subsamples can be used to reveal the uncertainty of the mean. In addition, the standard deviation (STD) is calculated as
(14)$$\sigma_{d,b}=\sqrt{\frac{\sum_{j=1}^{n}{\left({\bar{\Phi }}_{b,t_{i}}^{n,j}-{\bar{\Phi }}_{b,t_{i}}\right)}^{2}}{n}}$$
where ${\bar{\Phi }}_{b,t_{i}}$ refers to the daily, monthly or annual mean of the original total time series. The mean standard deviation can be further expressed as
(15)$$\bar{\sigma_{d,b}}=\frac{1}{N}\sum_{1}^{N}\sigma_{d,b}$$
where *N* refers to the number of the possible means in the total time series. For 2017, there are 365 days, so *N* is 365 when considering daily means and *N* is 12 when considering monthly means.

For the effect of different sampling methods, the error compared to the true mean for any given interval can be used. For a given sampling frequency of every *n* hours, the absolute error $\delta_{n,b,t_{i},j}$ and the average absolute error $\delta_{b,t_{i}}$ can be written as
(16)$$\delta_{n,b,t_{i},j}=\frac{\sum_{j=1}^{n}\left|\left({\bar{\Phi }}_{b,t_{i}}^{n,j}-{\bar{\Phi }}_{b,t_{i}}\right)\right|}{n}$$
(17)$$\delta_{b,t_{i}}=\frac{1}{N}\sum_{1}^{N}\delta_{n,b,t_{i},j}$$

To compare the structure of the time series and analyze the similarity between the original and subsampled time series, the correlation coefficient is introduced. The correlation coefficient between two time series provides a quantitative description for the similarity of two series. If a particular subsampled time series is more similar to the original time series than other subsampled time series, the similarity of this time series would be revealed by a high correlation coefficient [46,47,48]. Since the sampling frequency and start point are different, each subsampled time series is obtained from the same sample locations in the original time series. The correlation coefficients can be expressed as
(18)$$R_{n,b,t_{i},j}=\frac{COV({\bar{\Phi }}_{b,t_{i}}^{n,j},{\bar{\Phi }}_{b,t_{i}})}{\sqrt{{\bar{\Phi }}_{b,t_{i}}^{n,j}}\sqrt{{\bar{\Phi }}_{b,t_{i}}}}$$
where *COV* refers to the covariance and Φ is the variance.

#### 2.5.2. Discrete Fast Fourier Transforms (FFT)

The Fourier series and Fourier transform are common methods for analyzing periodic signals, and they have been widely used in digital signal processing and control systems. Because the positional relationship between the Sun, Earth and Moon and the Earth’s TOA scene in the field of view of the instrument change periodically, the irradiance of the MWFOV radiometer is a periodic discrete signal. Discrete Fast Fourier Transforms have been used to conduct frequency analyses on time series [49]. Through the Fourier transform method, the discrete signal in the time domain can be converted to the frequency domain, so that the frequency characteristics of the original discrete sequence in the time domain can be obtained. Therefore, this work uses the Discrete Fourier Transform (DFT) method with the number of sampling points in the frequency domain equal to the length of the discrete signal to analyze the periodical discrete irradiance obtained from the simulation. The function FFT provided by MATLAB software is used to complete Discrete Fourier Transform and, for the input irradiance discrete sequence *x*, the function FFT implements the relationships [50]:(19)$$Y(k)=\sum_{j=1}^{n}x(j)W_{n}^{\left(j-1)(k-1\right)}$$
(20)$$W_{n}=e^{-j2\pi /n}$$
where *n* is the length of discrete sequences *x* and *Y*. *W_n_* is one of *n* roots of unity, and *j* is the imaginary unit.

## 3. Results

In this section, the original and subsampling irradiance time series for the MWFOV radiometer produced by the CERES SYN1deg Ed4A are analyzed, and a discussion about the results is presented.

### 3.1. The Original Time Series

Based on the model constructed in Section 2, Figure 6a,c present the original shortwave (SW), longwave (LW) and the narrow ”window” (NW) channel irradiance time series on October in 2017 and for the entire year of 2017, respectively. The results show that the SW irradiance ranges from about 0.00 to 94.55 mW∙m^−2^, while for the LW and NW irradiance, the range is 58.05~86.92 and 14.09~25.79 mW∙m^−2^. The time series in Figure 6a,c also serve to demonstrate how the irradiance varies on hourly, daily and monthly time scales. In more detail, in October 2017, there is generally a sudden increase in outgoing shortwave radiation (OSR) with a peak on or around 18 October UTC. After the peak occurs, the OSR decreases to a monthly low at around 4 November UTC, and the value is zero. For the outgoing longwave radiation (OLR), the time of the peak value is around 6 October to 8 October UTC, and the low value is around 25 October to 26 October UTC. The latitude of the sub-Sun and sub-MWFOV point on Earth’s TOA over the Moon’s orbital period and one year are shown in Figure 6b,d, respectively. The positive latitude means that the sub-point is in the Earth’s Northern Hemisphere, and the negative latitude means the sub-point is in the Earth’s Southern Hemisphere. The variation of OLR has a strong correlation with the change trend of the latitude of the sub-MWFOV point, and LW irradiance is greater when the sub-MWFOV point is located in the Northern Hemisphere. As the variation of OSR depends on the geometric relationship of the Sun–Moon–Earth system, the OSR has a weak correlation with the change of the latitude of the sub-MWFOV point (Figure 6a,b). For a certain day, the lowest values of OSR generally coincide with the Sun being over the Pacific Ocean, and the irradiance will increase when a large land area comes into view. Then, tail-off occurs as a large ocean comes into the field of view, and different weather patterns and cloud locations cause differences from day to day. A comparison of Figure 6c,d indicates that the latitude variation of the sub-MWFOV point is the main reason for the periodic monthly oscillation of OLR and OSR. The change of the sub-Sun point latitude is the main reason for the peak yearly irradiance, and the time of the annual peak often corresponds to summer in the Northern Hemisphere. For longer periods, such as 5 years, 10 years and 20 years, LW and SW EPI have similar changes.

### 3.2. Temporal Sampling Uncertainties

Different sampling intervals and different sampling times will cause different uncertainties of irradiance. The irradiance time series of OLR and OSR from 15 October to 17 October in 2017 are shown in Figure 7, and the curves present the original and subsampled time series obtained with the same starting point and different sampling frequencies. The original time series, limited by the temporal resolution of the CERES data product, can serve to demonstrate how the irradiance varies on daily time scales (Figure 7). The variation of the land-to-sea area ratio and the MWFOV-radiometer-viewed area illuminated by the Sun causes the irradiance to change significantly throughout the day. Moreover, as the sampling frequency becomes increasingly coarser, more details of irradiance time series are lost. For example, while the sampling interval is 4 h, a loss of peak information will occur. In addition, a coarser sampling frequency will cause a significant change in the real shape of irradiance, meaning that the obtained data lose a great deal of high-frequency information.

The mean standard deviation (STD) of irradiance refers to the mean of the standard deviation of the difference between original and subsampled time series over all possible subsamples, so the value reveals the uncertainty of the sampling intervals. The mean daily STD of OLR, OSR and OLR–NW for different sampling intervals is shown in Figure 8. It is obvious that the mean STD of OSR is larger than OLR and the outgoing longwave radiation of the “window” channel (OLR-NW) under the same sampling interval. Since the mean STD is calculated by averaging the different daily subsamples, the range of variation is not very large, but it can reflect the overall sequence characteristics. The results in Figure 8 show that the mean daily STD of OSR ranges from about 2.00 to 2.12 mW∙m^−2^, while for the OLR and OLR-NW, the maximum value was about 1.30 and 0.55mW∙m^−2^, and the minimum was about 1.14 and 0.48 mW∙m^−2^, respectively. In addition, compared with the OSR, the OLR and OLR-NW have higher stability under the same sampling intervals. It should be noted that the mean daily STD does not always rise with the increase in sampling intervals. When the sampling interval is less than 8 h, the mean daily STD keeps increasing, while it starts to decrease when the sampling interval is greater than 8 h. The main reason for this is the daily cycle of irradiance and the use of average values. When the sampling interval is 8 h, the subsample time series include two sets of observations, each with three values. However, only three measurements per day will result in the loss of a great deal of information with a frequency higher than 8 h. When the sampling interval changes from 1 h to 8 h, the relative changes of mean STD are 6.5% (OSR), 12.3% (OLR) and 12.8% (OLR-NW). In addition, when the sampling interval is 1 h, the mean daily STD has the lowest value.

The mean daily STD can only reflect the long-term, overall features of irradiance, but cannot show the variations in irradiance on different time scales well. To investigate the temporal sampling uncertainties under different time scales, Figure 9, Figure 10 and Figure 11 present the daily mean and daily STD of irradiance with the sampling intervals of 2, 4, 6 and 8 h in October 2017, and the error bars reveal the mean daily standard STD of the subsampled time series. The vertical scaling for each figure is kept fixed, but the horizontal scale is not. The results in Figure 9, Figure 10 and Figure 11 show that the daily mean and daily STD of irradiance vary with the working band (SW, LW or NW) and the sampling intervals. There is no significant variation for the daily mean under different sampling intervals, but the variation ranges and the duration of drastic changes for OSR are greater than the OLR and OLR-NW (Figure 9, Figure 10 and Figure 11f). In more details, the daily STD peak of irradiance varies with the month, like the STD peak of SW occurring on or around 15 October to 19 October UTC, but for the other months, the time will change. In addition, the maximum of daily STD is around 2–4 times of the minimum; for example, the maximum of daily STD is 2.4 mW∙m^−2^ and the minimum is around 0.6 mW∙m^−2^ for the OSR. It is obvious that the difference of daily STD under different sampling intervals is different, as shown in Figure 9, Figure 10 and Figure 11f, and it varies with time. For example, the difference for OLR between 2 h and 6 h time series is 0.45 mW∙m^−2^ on 19 October 2017, while for 28 October 2017, it is 0.05 mW∙m^−2^ (Figure 11f).

The daily STD for different sampling intervals in 2017 is presented in Figure 12. Figure 12d is the original time series, and (a), (b) and (c) refer to the difference between the subsampled time series with the sampling intervals of 2, 4 and 6 h and the original time series. Compared to the mean daily STD (see Figure 8), the daily STD is periodic, and there is generally a sudden increase in daily STD with a peak at the end of each month to the beginning of the next month. After the peak occurs, the lowest values come at around mid-month. For the sampling interval of 1 h in 2017, the variation ranges of OSR, OLR, OLR-NW and OSR + OLR for daily STD are 0.00~5.32, 0.33~2.33, 0.12~1.10 and 0.38~4.98 mW∙m^−2^, respectively. The OSR + OLR is the sum of OSR and OLR. For a longer period from March 2000 to December 2020, the daily STD variation range will be further expanded, and the ranges of OSR, OLR, OLR-NW and OSR + OLR are 0.00~7.33, 0.08~3.07, 0.06~1.25 and 0.08~7.27 mW∙m^−2^, respectively. With an increase in the sampling interval from 1 h to 6 h, it is obvious that the uncertainty increases, and the change in deviation exhibits a positive and negative alternating phenomenon. For the OLR and OLR-NW cases, the daily STD absolute error between the 1 h sampling and 2 h sampling intervals is less than 0.05 mW∙m^−2^, and the relative deviation is less than 2.20% in 2017, while the deviation for OSR and OSR + OLR is less than 0.12 mW∙m^−2^ and the relative error is less than 2.24%. However, when the sampling interval is 6 h, the absolute deviations in 2017 are less than 0.62 (OSR), 0.30 (OLR), 0.14 (OLR-NW) and 0.58 (OSR + OLR) mW∙m^−2^, respectively, and the relative deviations are up to 26% (OSR), 12.90% (OLR), 12.87% (OLR-NW) and 36% (OSR + OLR), respectively.

Figure 13 presents the statistical histogram and normal distribution curve for the daily STD of irradiance for OSR, OLR and OLR-NW from March 2000 to December 2020. For the variables that obey the normal distribution, the “3-sigma” principle states that the probability of a variable falling outside the interval between *mu* − 3-sigma and *mu* + 3-sigma is less than 0.3%. The *mu* and sigma are the mean and the standard deviation. The statistical histogram shows that the normal characteristics of OLR and OLR-NW are more significant than OSR and OSR + OLR; that is, the symmetry of OLR and OLR-NW is stronger. The results in Figure 13 show that the *mu* has a slight variation for daily STD with the increase in sampling intervals. For the OSR, the *mu* is 2.00 under the sampling interval of 1 h, and it is 2.13 for the 6 h interval. Similarly, the *mu* of OLR is 1.15 in the 1 h interval and 1.28 in the 6 h interval, respectively. Since the total radiation is mainly controlled by OSR, the *mu* of OSR + OLR is close to OSR, and the deviation does not exceed 0.05 mW∙m^−2^.

Then, we calculated the monthly mean and STD of irradiance with the 1 h sampling interval during the period of March 2000 to December 2020, as shown in Figure 14. By comparing Figure 6, Figure 9, Figure 10 and Figure 11, the results show that as the analysis span increases from day, to month, to year, the mean and standard deviation become more stable, as shown in Table 2. In addition, from the monthly and annual means, it can be seen that the increase in the analysis span will weaken the periodicity, and the maximum and minimum values influence the drastic changes in the period, such as OSR in Figure 14. Based on the mean, considering the standard deviation, it is obvious that the standard deviation of OSR is approximately 4 times that of OLR and 12 times that of OLR-NW.

As anticipated, the uncertainty increases with the sample interval, and the STD will vary as the analysis span increases from day to year. Figure 15, constructed like Figure 12, shows the monthly STD difference of irradiance between the 1 h sampling interval time series and 2 h, 4 h, 6 h and 8 h time series. Similar to Figure 12, the STD of the difference between the original and the subsampled time series (2 h, 4 h, 6 h and 8 h) are larger for OSR and the sum of OSR plus OLR, because the OSR is always associated with smaller spatial and temporal scales. When using a sampling frequency of 2 h, the monthly STD of difference does not exceed 0.02 mW∙m^−2^. When increasing the sampling intervals from 1 h to 2 h, 2 h to 4 h, 4 h to 6 h or 6 h to 8 h, the monthly STD of the difference between the original and the subsampled time series increases by about 0.03 to 0.04. However, it is worth noting that even if the sampling frequency is 8 h, the monthly maximum STD of difference is still less than 0.14, 0.04, 0.012 and 0.018 mW∙m^−2^ for OSR, OLR, OLR-NW and OSR + OLR, respectively.

### 3.3. The Effect of Sampling Intervals and Sampling Time

For a given sampling frequency, the average absolute error of irradiance for OLR and OSR in 2017 is shown in Figure 16, and the vertical scale is kept fixed within each of the two sub-panels. It is obvious that when increasing the sampling intervals from 1 h to 2 h, to 4 h and to 6 h, the average absolute error between the original and subsamples keeps increasing and the error varies with the time. From the perspective of amplitude, as the intervals increase, the average absolute error will fluctuate more violently, and there will be a period of the most violent fluctuation every month, as in the period from October 17 to October 19 UTC for the OLR (Figure 16a). For the 1 h sampling frequency of OLR, the average absolute error ranges from about −1.17 to 1.24 mW∙m^−2^, while for the 6 h sampling frequency, the minimum value was about −3.33 mW∙m^−2^ and the maximum was about 5.38 mW∙m^−2^. For shortwave radiation, when using a sampling frequency of 1 h, the average absolute error is relatively small—generally between −4.65~4.13 mW∙m^−2^. However, for the 6 h sampling interval, the variation ranges of average absolute error are −8.94~13.21 mW∙m^−2^—approximately 2.5 times 1 h.

The statistical histogram for the average absolute error of irradiance for OLR, OSR and OLR-NW from March 2000 to December 2020 are presented in Figure 17, Figure 18 and Figure 19. Note that the vertical scaling is the error, and the y-axis corresponds to the number of measurements. The statistical results show that the average absolute error of irradiance presents a normal distribution, and the shape of the normal distribution shows that the average absolute error for OSR is larger. The “3-sigma” principle states that the probability of a variable falling outside the interval between *mu* − 3-sigma and *mu* + 3-sigma, where *mu* and sigma are the mean and the standard deviation, is less than 0.3%. As the range (*mu* − 3-sigma, *mu* + 3-sigma) includes 99.7% of the possible error values, the value of “3-sigma” can be used to measure the margin of error. For the 1 h sampling frequency of OSR, OLR and OLR-NW, the “3-sigma” values are 1.82, 1.20 and 0.60 mW∙m^−2^, respectively, but when using the 6 h sampling intervals, the “3-sigma” values are 8.25, 4.68 and 2.01 mW∙m^−2^, respectively. When the sampling interval increases by six times, the deviation increases by about three times. To accurately identify the high-frequency variation in irradiance for the MWFOV radiometer, not only a high radiation resolution but also high stability is required. Therefore, the statistical properties of the absolute deviation will help us to understand the stability of irradiance.

As the time series of measurements vary with the sampling intervals and start time, it is important to evaluate the sampling similarity with the correlations of daily mean irradiance between original and subsampled time series. Table 3 shows the correlations of daily means for OSR, OLR, OLR-NW and OSR + OLR under different sampling intervals and different start times. The results in Table 3 show that the correlation coefficients are generally close to one, and a slight decrease in terms of the daily mean correlation coefficients for different sampling frequencies is found. For 2 h sampling frequencies, the correlation is 1, while for the 6 h, the correlation coefficients are close to 0.9997 for OSR, 0.9996 for OLR, 0.9992 for OLR-NW and 0.9996 for OLR + OSR, respectively. For the 8 h sampling frequency, with the increase in the start time, the correlation decreases from 0.9996 at the first hour to 0.9986 at the eighth hour, and the change for OLR is not obvious. Moreover, the correlations of the time series are very similar for all starting points when the sampling interval is less than 6 h and for the all-sampling interval. In addition, the correlations between the original and subsampled time series at the start times of 3:00 and 4:00 are higher overall than other time series. In summary, it is better to have higher correlations for a certain sampling interval, and it is obvious that the influence of sampling intervals and start times will be smaller when the sampling interval is smaller than 4 h, as shown in Table 3.

### 3.4. The Effect of Sampling Intervals on Coverage Ability

From the perspective of spatial coverage ability, the MWFOV-radiometer-viewed region will vary with the sampling interval, and when the sampling interval is less than 12 h, the adjacent observations have part of the same scene. Figure 20 presents the SW and LW coverage images of different sampling intervals per day. The figures are obtained by plotting the MWFOV-radiometer-viewed area of different sampling sequences on 180 × 360 grids (1° × 1°) and the number of visible grids is normalized, i.e., the ratio 1 in Figure 20a reveals that these regions on the Earth’s TOA can be seen 12 times in one day, or 2 times for the sampling interval of 6 h. As the sampling frequency decreases from every 1 h to 2 h, or every 2 h to 4 h, or every 4 h to 6 h, as shown in Figure 20, the overlapping area of two adjacent observations reduces, and the number of observations in many places per day is less than the maximum number of observations (the ratio of 24 h to the sampling interval). In addition, the decrease in sampling frequency will lead to a significant reduction in the number of observations per day for any given location. In addition, as the “window” and longwave channel is similar in the MWFOV-radiometer-viewed area, the coverage images of NW are not shown here.

Table 4 shows the SW and LW statistics of the area ratio for the different sampling times under different sampling frequencies. For the OSR, when the sampling interval is 1 h, 82.28% of the Earth’s surface grid area can be sampled more than nine times, and the sampling is relatively uniform and stable. However, for the sampling interval of 2 h, 65.17% of the area can only be sampled five times, and 12.37% can only be sampled six times. When the sampling interval is 6 h, 57.82% of the Earth’s surface area can only be sampled twice, and 33.08% of the area can only be sampled once. The OLR is not directly affected by solar radiation, so the coverage ability is better than SW radiation. The results in Table 4 show that for the 1-h sampling frequency of OLR, 97.52% of the Earth’s surface grid area will be sampled more than nine times, but for the sampling frequency of 2 h, 72.78% of the area can only be sampled six times and 4.76% can only be sampled seven times. When the sampling frequency decreases to 6 h, 89.38% of the Earth’s surface area can only be sampled twice per day. In summary, a small sampling interval will allow more observation opportunities for any given location. It should be noted that the coverage images (in Figure 20) and area ratio (in Table 4) are acquired at the maximum moment of OSR (see Figure 6), so the grid area ratio sampled more than nine times with a 1-h sampling interval can reach up to 82%. When the sampling interval is better than 1 h, for a given location, higher-resolution observation data will be obtained, and these data will support for the study of high-frequency, large-scale changes to Earth’s energy.

### 3.5. Frequency Analysis of Irradiance

As shown in Figure 6, the original time series show a diurnal and monthly cycle and numerous sub-diurnal modes of variability across a wide range of time scales, which indicates that the frequency analysis of these observations is warranted. In addition, high-quality data can always reflect their own internal characteristics, such as period and amplitude. Therefore, the period extraction and frequency analysis of discrete simulated irradiance data are important aspects of evaluating data quality. In this work, the DFT is used to analyze the period of discrete irradiance data. The relationship between magnitude and frequency for the original OSR and OLR irradiance time series from March 2000 to December 2020 is shown in Figure 21 and Figure 22, respectively. The vertical scaling is the normalized frequency with the 1-h sampling frequency, and the y-axis represents the magnitude of the Fourier series. The smaller the value of the frequency, the longer the corresponding period. The results show that the Discrete Fourier Transforms of the original irradiance time series in the frequency domain can reflect and extract the internal period or frequency characteristic for the discrete irradiance time series, and the magnitude and the number of amplitude peaks reflect the strength of the period. It is obvious that the main period of irradiance for the OSR and OLR, such as hours, days, months, and years, can be obtained by DFT, as shown in Figure 21 and Figure 22.

From the number of amplitude peaks, the OSR has a more complex period compared with the OLR, and the results in Figure 21 and Figure 22 are consistent with those in Figure 6. The 1-h period is mainly caused by the temporal resolution of the satellite data product and the variation in the scene of MWFOV-radiometer-viewed area. For the shortwave radiation, the appearance of the 1.04 h, 1.09 h, 6 h and 8 h periods is caused by the drastic daily changes of cloud, scene types and shortwave anisotropy, and the reason is the same for the 1.04 h period of the longwave radiation. The alternation of ocean and land in the radiometer’s field of view brings a period of 12.5 h, and the movement of the Moon around the Earth may have caused 10 day or 14.9 day and monthly periods. It should be noted that the diurnal cycle of irradiance is a major feature of the time series. For longer time spans, half-year and one-year periods will also be obtained by the Discrete Fourier Transform. It should be noted that the features with the periods of days, months and years could be extracted by the DFT for all the subsample time series with the sampling intervals of 2 h, 4 h, 6 h and 8 h, but as the sampling frequency increases, the information that can be extracted in irradiance will increase. However, there exists a difficulty in extracting the features on the minute scale because it has a great dependence on the temporal resolution.

## 4. Discussion

As a special platform, compared to artificial satellites, the Moon-based platform can provide a longer-term, continuous Earth radiation observation at a planetary hemispherical scale. The high orbital altitude of Moon, which is approximately 10 times that of geostationary orbit and 60 times that of a polar-orbiting-satellite, means that the reflected solar shortwave (SW) radiation and outgoing longwave infrared (LW) radiation from the Earth’s top of atmosphere can be measured by considering the Earth as a single pixel. A non-scanning Moon-based Wide Field-of-View (MWFOV) radiometer can provide a powerful supplement to the data of satellite platforms for the ERB. Limited in its own synchronous rotation, the nearside of the Moon always faces Earth. The rotation cycle of the Earth is 24 h; therefore, the nearside can sample the whole Earth in one day, which will be helpful to overcome the shortcomings of half of the Earth always being invisible from the Moon. Most of regions on the Earth’s TOA can be sampled 12 times in one day if the sampling interval is set to 60 min, and almost the entire earth can be observed in one day. Similarly, 48 temporal samples can be taken per day if the sampling interval is set to 15 min. The measurements of the Earth’s outgoing radiation for the MWFOV radiometer are closely related to the temporal sampling interval, and the sampling interval represents the time difference that the adjacent samplings have. A shorter sampling interval would result in more samples for a certain period, but the radiometer would need a more precise design and have a higher cost. The study of the effect of the temporal sampling interval on the irradiance for the MWFOV radiometer can optimize the sampling frequency setting of this instrument, and it also serves as a data support for optimizing the design of the instrument to obtain a high signal-to-noise ratio. Within this context, the lunar surface site 0° E 0° N is selected as the position of the radiometer, and the simulated original irradiance time series, used as the substitute for the truth, are obtained by utilization of the CER_SYN1deg-1Hour_Edition4 data products. By analyzing the simulated original and subsampling irradiance time series with the help of the defined metrics and Fourier transform, the effect of temporal sampling interval on the irradiance is finally studied.

Different sampling intervals will have different subsample time series. When the sampling interval is 1 h, the irradiance shows a significant change on hourly, daily and monthly time scales, as shown in Figure 6, but as the sampling frequency becomes increasingly coarser, more details of irradiance time series are lost (see Figure 7). The analysis of temporal sampling error can acquire the change characteristics of irradiance with time under different sampling methods. For the 1 h sampling frequency of OSR, OLR and OLR-NW, the “3-sigma” values are 1.82, 1.20 and 0.60 mW∙m^−2^, respectively, but when using the 6 h sampling intervals, the “3-sigma” values are 8.25, 4.68 and 2.01 mW∙m^−2^, respectively. This derived result will facilitate the ascertainment of the time response design and materials selection of the radiometer; in other words, in order to achieve the accurate detection of such a small amount of radiation, <0.5 or <1 mW∙m^−2^, with what kind of conditions should the time response characteristics of the radiometer be understood, and what kind of manufacturing material should be selected? Therefore, the choice of sampling interval will be of great importance for choosing optimal detection elements to obtain the highest SNR under a certain manufacturing budget.

Due to the limited sampling interval, the sampling times of the Earth’s surface area and the spatial coverage ability of the MWFOV radiometer will vary under different sampling frequencies. When the sampling interval is better than 1 h, for a given location, higher-resolution observation data will be acquired, and these data will promote the study of high-frequency, large-scale changes to Earth’s energy. In addition, the Discrete Fourier Transform (DFT) of discrete simulated irradiance data shows that the times series in the frequency domain can easily reflect the internal period or frequency of irradiance, such as the period of an hour, one day and a month. Recent work has also shown that the development of Earth System Model tuning for centennial-length simulations and the exploration of structural versus parametric error suggest that model constraints at high frequencies are warranted [51]. Therefore, the irradiance across time scales ranging from sub-diurnal to decadal from the MWFOV radiometer can be used for time-varying energetic constraints of the coupled inter-comparison project 5 model (CMIP5) and CMIP6 models [49].

As shown in Section 3.1, the order of magnitude of the entrance pupil irradiance for the MWFOV radiometer is 10^−2^ W∙m^−2^, and the magnitudes are not significantly different from the ERBE, CERES and GERB instruments. In addition, the magnitude of irradiance measured by the NISTAR on DSCOVR is also on the order of 10^−3^ W∙m^−2^, and the accuracy of the detector is 1.5% or better [11], providing a meaningful reference and verification for the realization of the MWFOV radiometer. When the sampling interval is better than 1 h, more observation data will be obtained for every Earth surface point, but if the radiation resolution of the radiometer cannot be further improved, two adjacent observations will be closer to the same value. If the minimum radiation resolution of the MWFOV radiometer is 1 mW∙m^−2^ with a high stability and the sampling interval is 1 h, two adjacent observations will have nearly the same value for OLR and OLR-NW, and for the OSR, 68% of the time, these will be the same (see Figure 17, Figure 18 and Figure 19). Therefore, in order to obtain more different and meaningful measurements, the radiation resolution of the MWFOV radiometer should be better than 0.5 mW∙m^−2^ in the future actual design, and the sampling interval should be less than 1 h. The achievement of this sensitivity design goal also requires the corresponding detection element and temperature control system of the MWFOV radiometer to meet certain requirements, such as accurate photoelectric equivalent measurement [34]. Based on the proved state-of-the-art spaceborne technologies, the current detector design technology could meet the requirements of the MWFOV radiometer’s design specification for irradiance [12,32,34,52,53].

In addition, SI traceability and accurate radiometric calibration for the MWFOV radiometer can be achieved by observing the cold and stable deep space background, using a dedicated stable calibration source and the mutual correction method of the established observation network. A more accurate radiometric calibration can also be achieved on the lunar surface based on the stable selenographic radiation environment. Indeed, the Moon has been used as a calibration reference source for low Earth orbit satellites, such as SeaWiFS onboard the OrbView-2 [54] and the MODIS onboard the Terra and Aqura satellites [55]. Since the early 1970s, great efforts have been made to deal with these data obtained on satellite platforms at different heights, such as the Earth Radiation Budget Satellite (ERBS) and the Clouds and the Earth’s Radiant Energy System (CERES); furthermore, corresponding complete methods for estimating the Earth’s radiation budget using satellite-based data have been developed. Therefore, the corresponding Moon-based data processing method can be developed on the basis of the processing method for the satellite-based data after being combined with the characteristics of the Moon-based platform [11,38,39].

It should be noted that this work is a study of the effect of the temporal sampling interval on the irradiance for a one-pixel Moon-based Wide Field-of-View (MWFOV) radiometer from the perspective of the simulations. Although the ADMs used were derived from the Nimbus 7 Scanner in the last century, it is a relatively mature angular dependency model for converting the radiance to a flux estimate, and this model is still being used in CERES data products, such as CER_ES4_Terra+Aqua_Edition4. Since the highest temporal resolution of CERES data products is 1 h, information for less than one hour is currently difficult to obtain through the method of this work, but its trend can be properly predicted. The specific design method for an MWFOV radiometer will be considered in the prototype design. Other factors related to the radiometer, such as structure design, material selection and retrieval algorithms, etc., which can also influence the quality of the observed data, are not considered. In addition, many countries around the world have been actively carrying out scientific explorations using the Moon as a scientific platform, such as the “Chang’E Project” and the Deep Space Gateway [56,57]. Therefore, the key technologies associated with the Moon-based platform will be verified in the following lunar exploration project.

## 5. Conclusions

Long-term, continuous measurements of the Earth’s outgoing radiation can be achieved by the non-scanning Moon-based Wide Field-of-View (MWFOV) radiometer, which will be a powerful data complement for analyzing the climate and predicting future global climate change. Based on the radiation transfer model, the simulated original and subsampled irradiance time series from March 2000 to December 2020 are obtained with the help of the highest-temporal-resolution CERES data products and ERBE angular distribution models (ADMs). By using the simulated irradiance time series, the defined metrics and the Fourier transform, the effect of the temporal sampling interval on the irradiance for a one-pixel MWFOV radiometer is analyzed. 

We first present the simulated original irradiance time series with the sampling interval of 1 h, and the result shows that the daily irradiance shows a strong hourly and daily oscillation, and the outgoing shortwave radiation (OSR) has a stronger daily variation compared to the outgoing longwave radiation (OLR) and the outgoing longwave radiation of the “window” channel (OLR-NW). In addition, the measurements of the MWFOV radiometer could reveal the variation of irradiance on hourly, daily and monthly time scales. The hourly measurements of irradiance can also reflect the variation of scene types in the MWFOV-viewed area, such as the ocean or land. Then, a set of metrics—the mean, the standard deviation (STD), average absolute error and the correlation coefficients—are used to analyze the effect of the temporal sampling interval on the irradiance. The results indicate that as the sampling frequency becomes increasingly coarser, more details of irradiance time series are lost. In the sensor’s field of view, the alternations of sea–land and day–night mean that the mean STD does not always rise with the increase in sampling intervals. The standard deviation at a certain sampling interval shows the characteristics of periodic variation. The decrease in sampling frequency from 1 h to 2 h or 2 h to 4 h will not cause a significant change in the mean daily STD, and the variation does not exceed 0.5 mW∙m^−2^. As the analysis span increases from day to month to year, the mean and standard deviation become more stable.

The metric of the average absolute error of irradiance can reveal the relationships between the magnitude of the maximum absolute errors and the sampling intervals. The statistical analysis for the average absolute error of irradiance reveals that the sampling frequency of 1 h has a minimum absolute error, and more useful data can be obtained. When the radiation resolution is better than 0.5 mW∙m^−2^ under the accuracy of 1% or better, more different and meaningful measurements will be obtained. In more details, the metric of correlation coefficients—a measure of the similarity of the two time series—show that a sampling interval smaller than 4 h results in a higher similarity (>0.9998) to the original time series, and the correlations of different start times at the same interval are very similar. 

From the perspective of spatial coverage ability, the results of the sampling times of the Earth’s TOA area show that when the sampling interval is better than 1 h, for a given location, higher-resolution observation data will be obtained, which can ensure that 97% of the surface area can be sampled more than nine times per day for longwave radiation. In addition, the Discrete Fourier Transforms of irradiance time series can reflect the internal period or frequency of irradiance, such as the period of an hour, one day and a month. As the sampling frequency increases, the extractable useful information of measurements will increase. The findings of this work will facilitate the optimization of the design of the radiometer to obtain a high signal-to-noise ratio and the selection of the sampling mode under a certain manufacturing cost and error limit. The other issues, such as structure design and retrieval algorithms, related to the radiometer will be discussed in a subsequent paper.

## Figures and Tables

**Figure 1 sensors-22-01581-f001:**
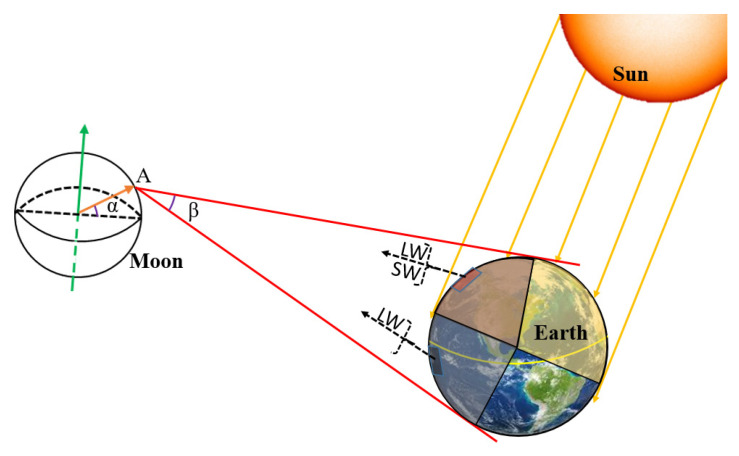
Geometrical relationship between the Earth, the Sun and the one-pixel Moon-based Wide Field-of-View radiometer.

**Figure 2 sensors-22-01581-f002:**
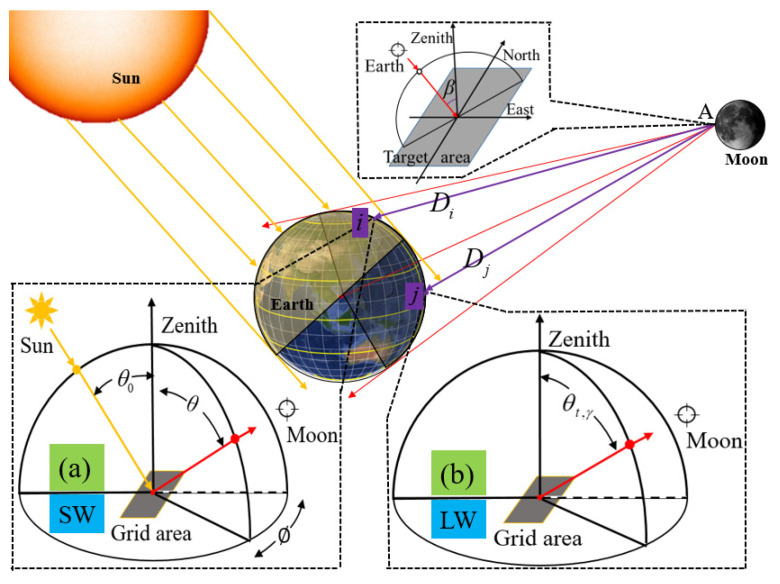
A sketch to illustrate the radiation transfer and the angular coordinate system for (**a**) SW and (**b**) LW angular distribution models of ERBE.

**Figure 3 sensors-22-01581-f003:**
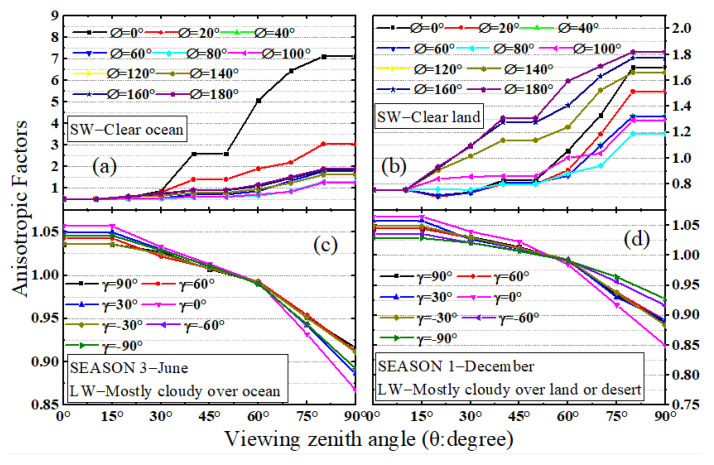
The ERBE TOA anisotropic factors for (**a**): SW clear ocean, (**b**): SW clear land, (**c**): LW mostly cloudy over ocean in June of season 3 and (**d**): LW mostly cloudy over land or desert, as a function of the viewing zenith angle θ (see Figure 2 for the detailed angle relationship).

**Figure 4 sensors-22-01581-f004:**
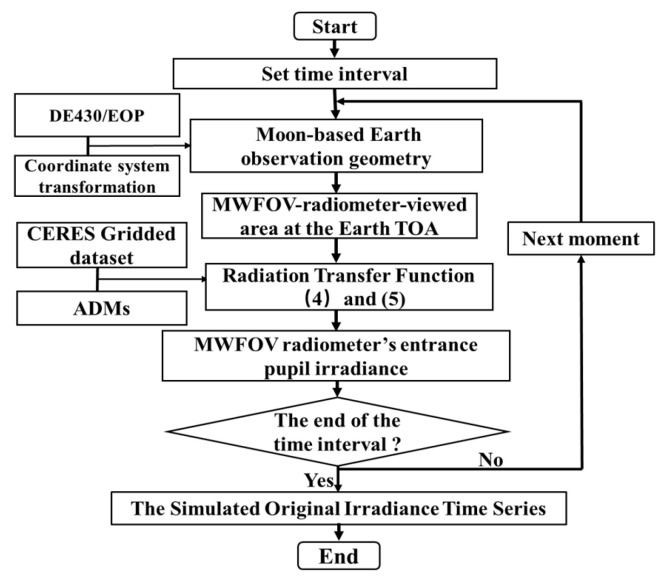
Flowchart for the calculation of irradiance. ADMs: angular distribution models.

**Figure 5 sensors-22-01581-f005:**
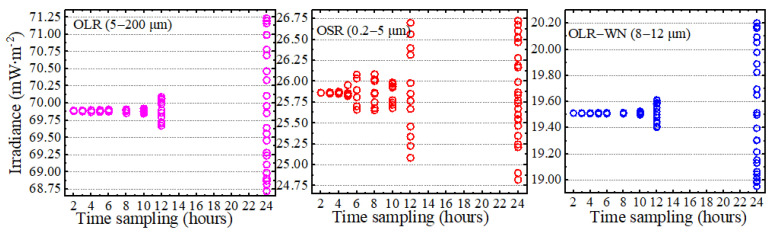
The monthly means in October 2017 for all of the realizations from the various sampling intervals and start point.

**Figure 6 sensors-22-01581-f006:**
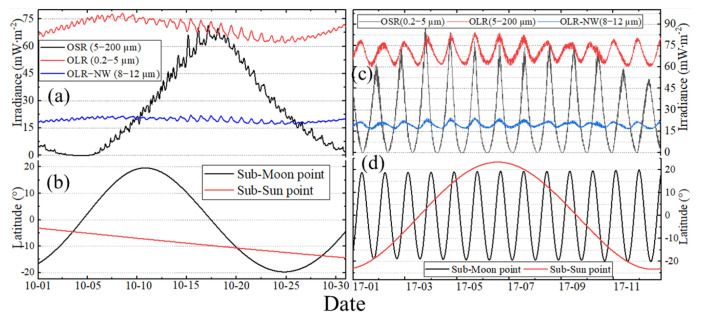
The original times series of the instrument’s entrance pupil irradiance and the latitude of sub-Sun or sub-Moon point on Earth. (**a**,**c**) are the irradiance in October in 2017 and for the whole of 2017, and (**b**,**d**) is the latitude of the nadir.

**Figure 7 sensors-22-01581-f007:**
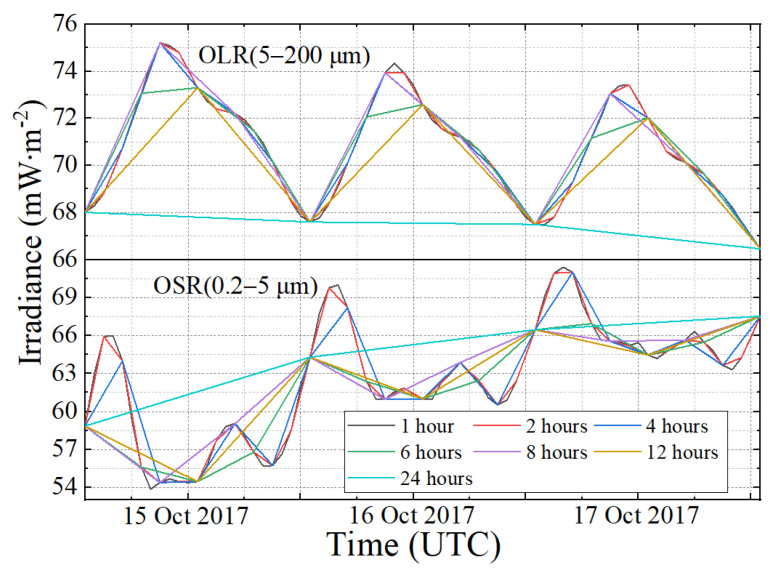
The irradiance simulated time series of outgoing longwave radiation and outgoing shortwave radiation for a three-day period from 15 October to 17 October in 2017. Colored curves show single realizations from the various sampling intervals.

**Figure 8 sensors-22-01581-f008:**
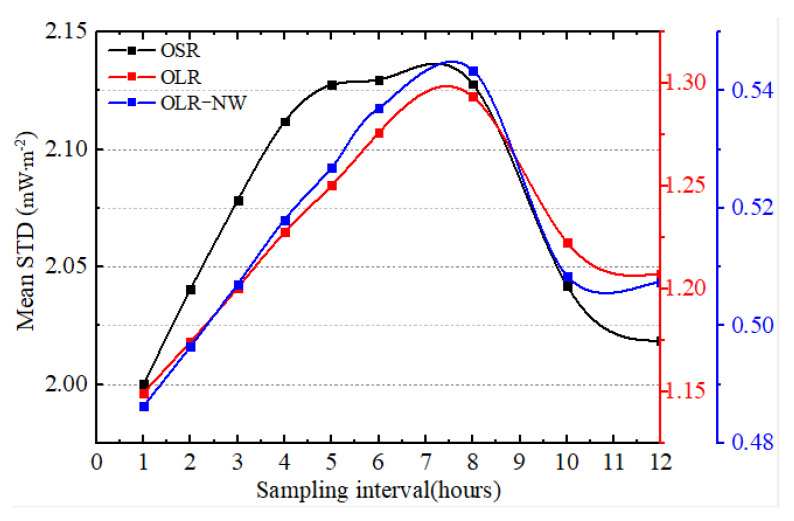
The mean daily STD of irradiance for different sampling intervals of OLR, OSR and OLR–NW. (The x-axis shows the sampling intervals—i.e., the sampling interval of 2 h refers to 2 sets of observations, each with 12 values).

**Figure 9 sensors-22-01581-f009:**
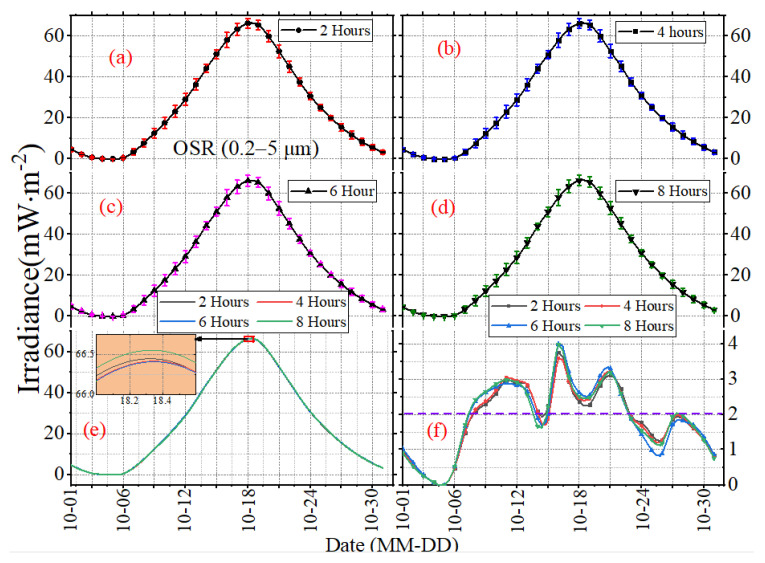
The daily mean and STD of SW irradiance with sampling intervals of (**a**) 2 h, (**b**) 4 h, (**c**) 6 h and (**d**) 8 h in October 2017. The curves in (**e**) are the mean values, and the curves in (**f**) are the standard deviation.

**Figure 10 sensors-22-01581-f010:**
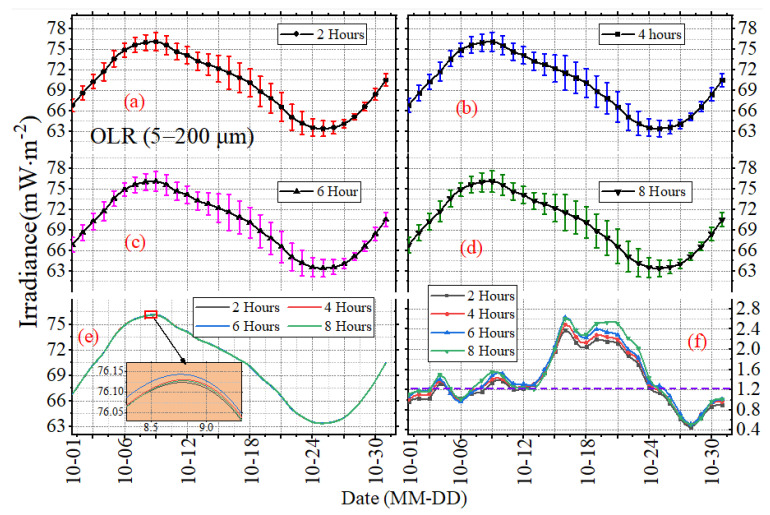
The daily mean and STD of LW irradiance with sampling intervals of (**a**) 2 h, (**b**) 4 h, (**c**) 6 h and (**d**) 8 h in October 2017. The curves in (**e**) are the mean values, and the curves in (**f**) are the standard deviation.

**Figure 11 sensors-22-01581-f011:**
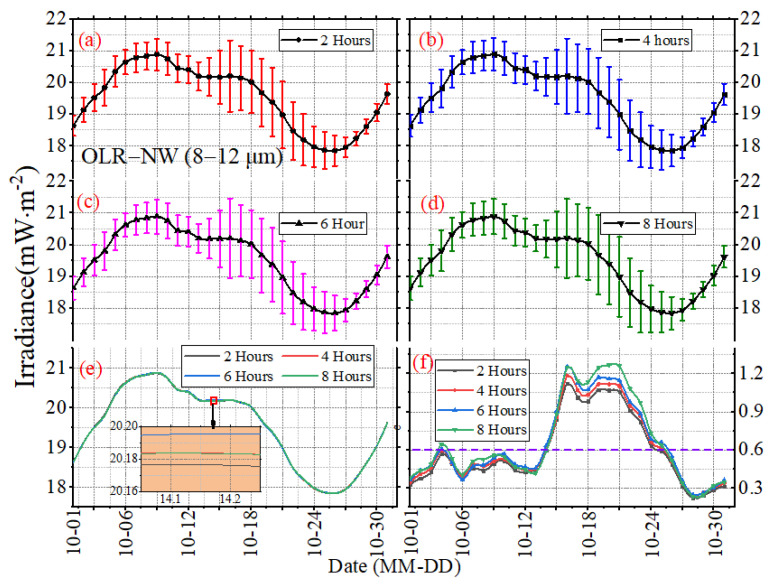
The daily mean and STD of LW–NW irradiance with the sampling intervals of (**a**) 2 h, (**b**) 4 h, (**c**) 6 h and (**d**) 8 h in October 2017. The curves in (**e**) are the mean values, and the curves in (**f**) are the standard deviation.

**Figure 12 sensors-22-01581-f012:**
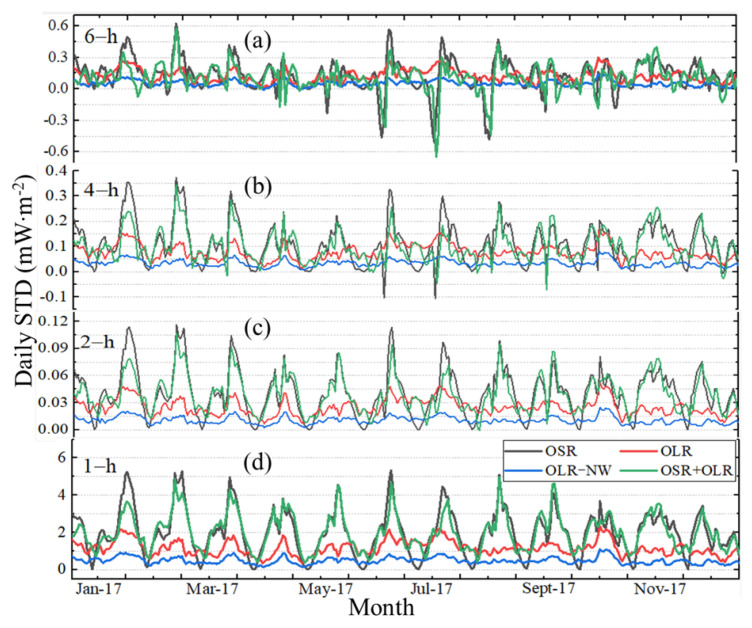
The daily STD of irradiance under different sampling intervals in 2017. (**d**) The original time series with the sampling interval of 1 h. (**a**–**c**) refer to the difference between 2 h, 4 h, 6 h and 1 h intervals.

**Figure 13 sensors-22-01581-f013:**
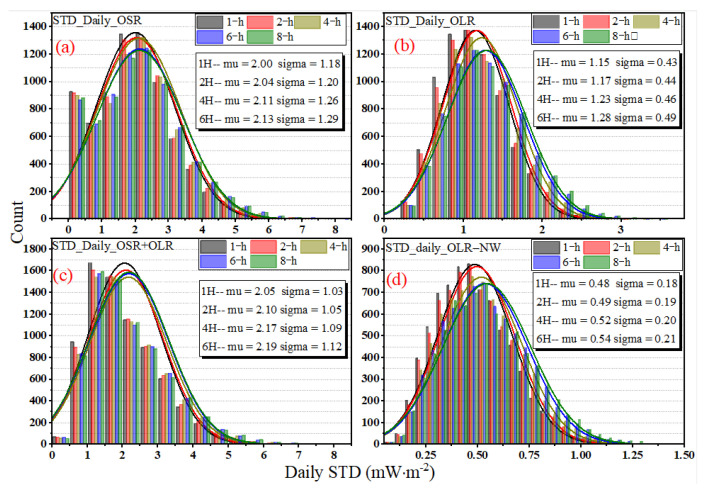
The statistical histogram and normal distribution curve for the daily STD of irradiance for OSR, OLR and OLR-NW from March 2000 to December 2020. The normal distribution curves are plotted according to normal distribution functions with the given mu and sigma values, and the mu and sigma values are showed in the subfigures.

**Figure 14 sensors-22-01581-f014:**
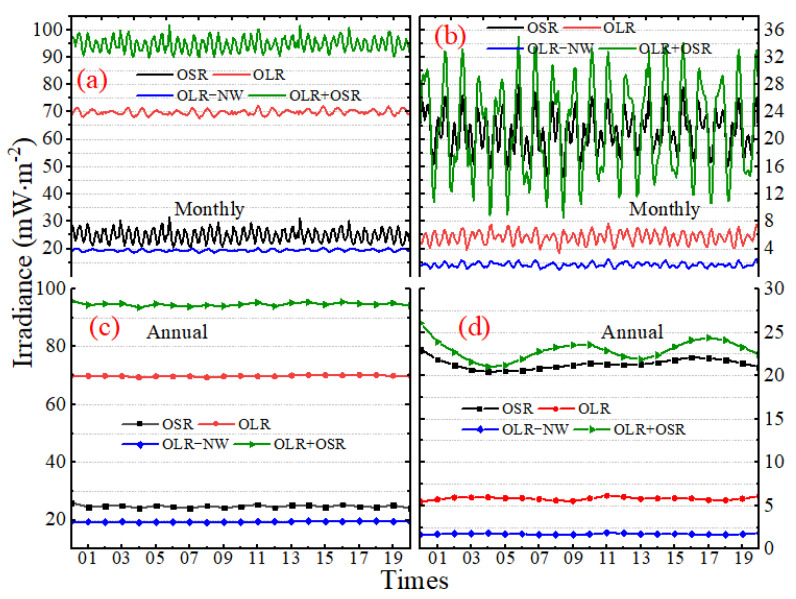
The mean and STD of irradiance with the 1 h sampling interval for (**a**,**b**) monthly; and (**c**,**d**) annual intervals during the period March 2000 to December 2020.

**Figure 15 sensors-22-01581-f015:**
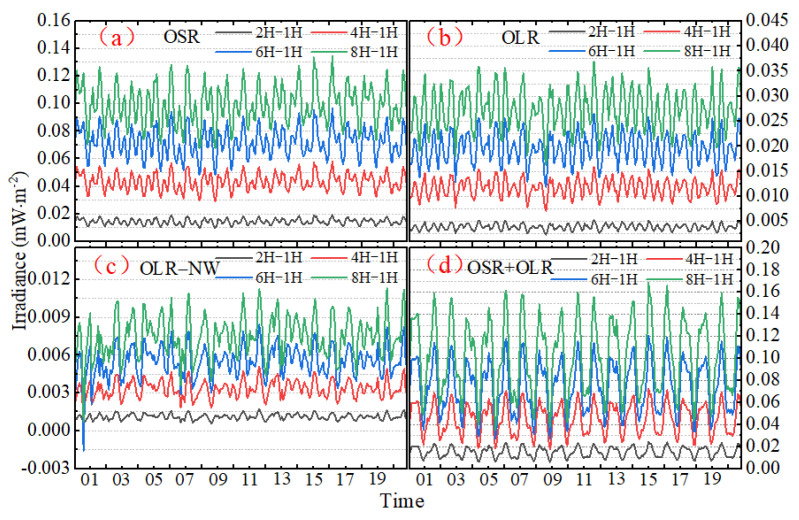
The monthly STD difference of irradiance between the 1 h sampling interval and 2 h, 4 h, 6 h and 8 h for (**a**) OSR; (**b**) OLR; (**c**) OLR-NW; (**d**) OSR + OLR.

**Figure 16 sensors-22-01581-f016:**
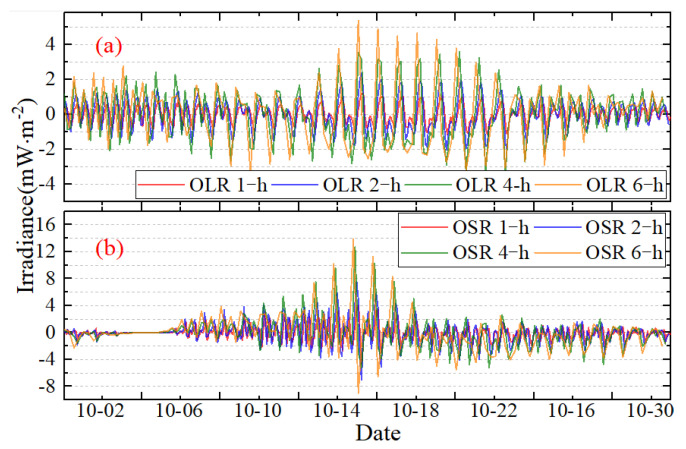
The average absolute error of irradiance for OLR (**a**) and OSR (**b**) in 2017.

**Figure 17 sensors-22-01581-f017:**
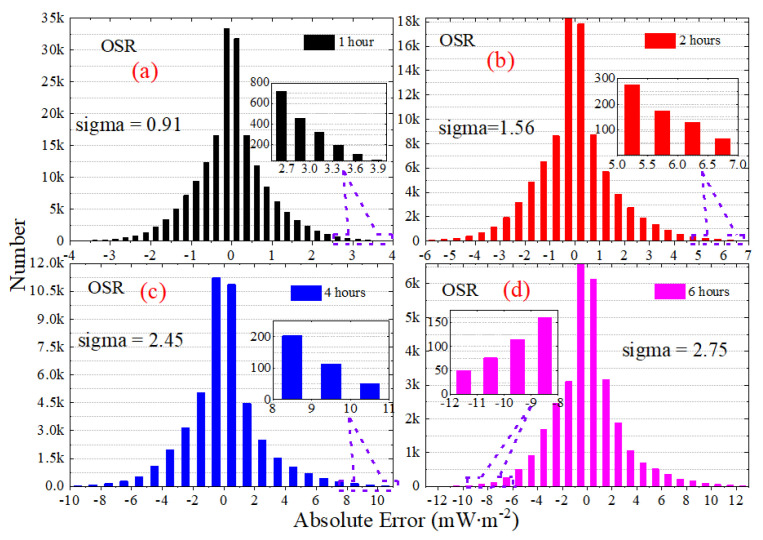
The statistical histogram for the average absolute error of irradiance for OSR from March 2000 to December 2020. The subfigures is the partial enlarged view for the explanation of OSR.

**Figure 18 sensors-22-01581-f018:**
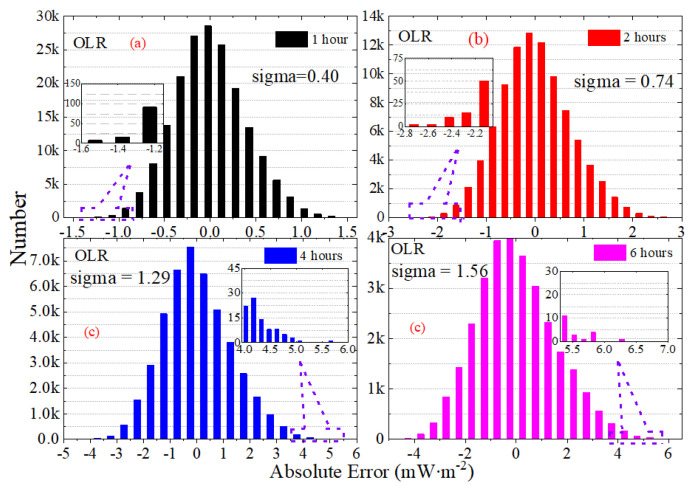
The statistical histogram for the average absolute error of irradiance for OLR from March 2000 to December 2020. The subfigures is the partial enlarged view for the explanation of OLR.

**Figure 19 sensors-22-01581-f019:**
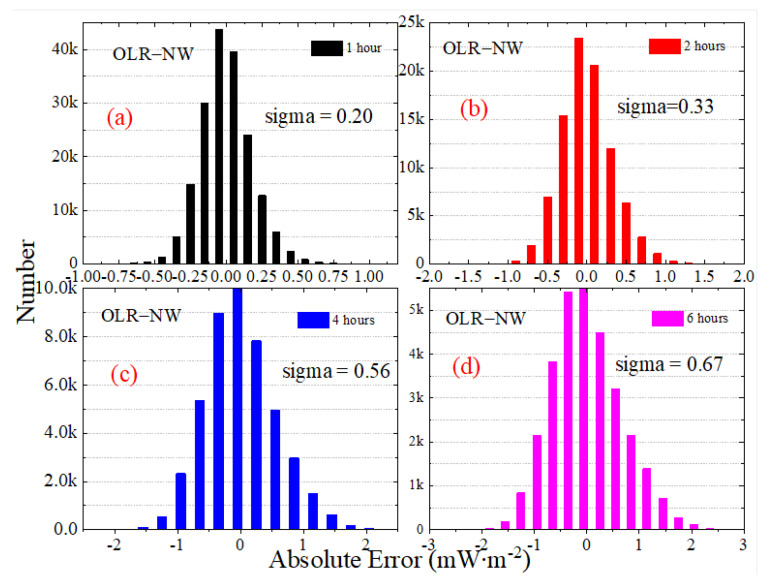
The statistical histogram for the average absolute error of irradiance for OLR-NW from March 2000 to December 2020. The subfigures is the partial enlarged view for the explanation of OLR−NW.

**Figure 20 sensors-22-01581-f020:**
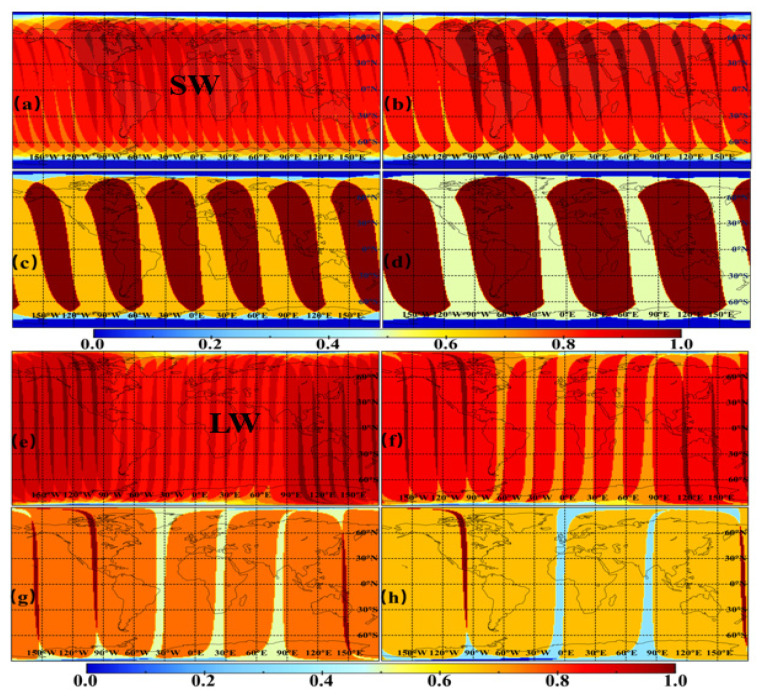
The SW and LW coverage images of different sampling intervals for the MWFOV radiometer on 17 October 2017 and 5 October 2017, respectively. (**a**,**e**), (**b**,**f**), (**c**,**g**) and (**d**,**h**) represent the 1, 2, 4 and 6 h sampling intervals. The deep red part is the visible region for every observation of the MWFOV radiometer, and the number of visible grids is normalized.

**Figure 21 sensors-22-01581-f021:**
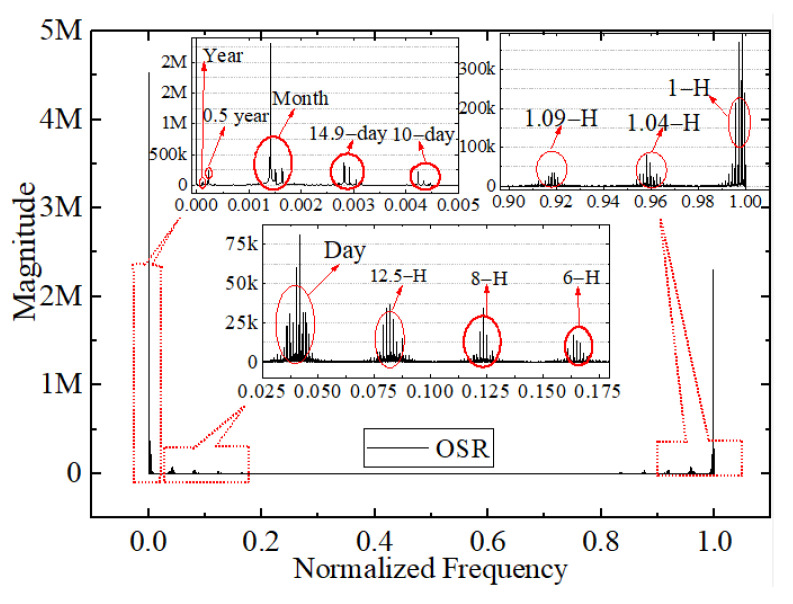
The relationship between magnitude and normalized frequency for the original OSR simulated irradiance time series from March 2000 to December 2020.

**Figure 22 sensors-22-01581-f022:**
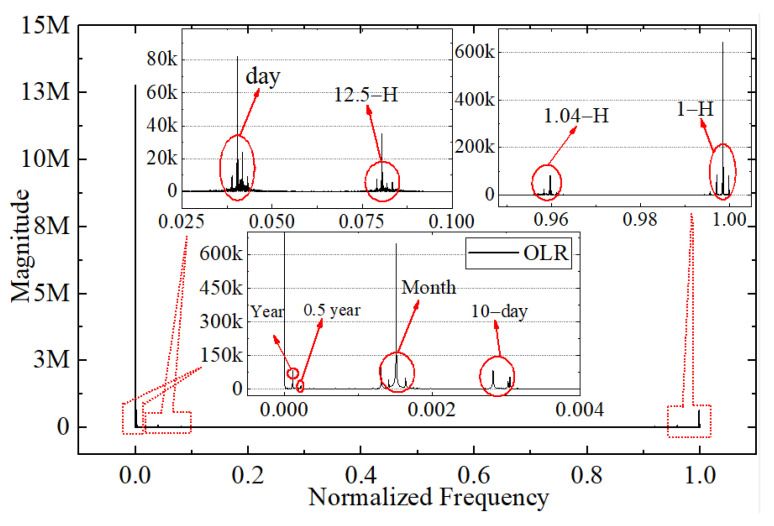
The relationship between magnitude and normalized frequency for the original OLR irradiance simulated time series from March 2000 to December 2020.

**Table 1 sensors-22-01581-t001:** The scene types of angular distribution models (ADMs) for ERBE.

Scene Type Bin	Scene	Cloud Fraction Range
1	Clear Ocean	0.00−0.05
2	Clear Land	0.00−0.05
3	Clear Snow	0.00−0.05
4	Clear Desert	0.00−0.05
5	Clear Land–Ocean Mix (Coastal)	0.00−0.05
6	Partly Cloudy Over Ocean	0.05−0.50
7	Partly Cloudy Over Land or Desert	0.05−0.50
8	Partly Cloudy Over Land–Ocean Mix	0.05−0.50
9	Mostly Cloudy Over Ocean	0.50−0.95
10	Mostly Cloudy Over Land or Desert	0.50−0.95
11	Mostly Cloudy Over Land–Ocean Mix	0.50−0.95
12	Overcast	0.95−1.00

**Table 2 sensors-22-01581-t002:** The monthly mean and STD, and annual mean and STD during the period March 2000 to December 2020 (units: mW∙m^−2^).

		OSR	OLR	OLR-NW	OSR + OLR
Monthly	Mean	20.54~31.45	67.59~72.24	18.45~20.34	89.75~101.53
	STD	14.46~27.95	3.40~7.63	0.90~2.53	8.57~34.93
Annual	Mean	24.10~26.03	69.4~70.23	19.25~19.63	93.61~95.93
	STD	20.47~23.06	5.46~6.15	1.65~1.86	21.03~26.08

**Table 3 sensors-22-01581-t003:** The correlations of daily mean OSR, OLR, OLR − NW and OSR + OLR for different start times under the sampling intervals of 2 h, 4 h, 6 h and 8 h.

	Interval	Start Time							
		1	2	3	4	5	6	7	8
OSR	2 h	1	1	×	×	×	×	×	×
	4 h	0.9999	0.9999	0.9999	0.9999	×	×	×	×
	6 h	0.9994	0.9997	0.9999	0.9998	0.9997	0.9997	×	×
	8 h	0.9996	0.9996	0.9997	0.9996	0.9996	0.9996	0.9990	0.9986
OLR	2 h	1	1	×	×	×	×	×	×
	4 h	0.9998	0.9999	0.9999	0.9999	×	×	×	×
	6 h	0.9995	0.9997	0.9999	0.9999	0.9997	0.9995	×	×
	8 h	0.9992	0.9995	0.9998	0.9998	0.9998	0.9997	0.9991	0.9992
OLR − NW	2 h	1	1	×	×	×	×	×	×
	4 h	0.9998	0.9998	0.9999	0.9997	×	×	×	×
	6 h	0.9992	0.9992	0.9993	0.9995	0.9992	0.9992	×	×
	8 h	0.9996	0.9990	0.9993	0.9990	0.9986	0.9986	0.9988	0.9986
OLR + OSR	2 h	1	1	×	×	×	×	×	×
	4 h	0.9999	0.9999	0.9999	0.9999	×	×	×	×
	6 h	0.9994	0.9995	0.9999	0.9998	0.9996	0.9997	×	×
	8 h	0.9996	0.9996	0.9998	0.9997	0.9996	0.9996	0.9989	0.9987

**Table 4 sensors-22-01581-t004:** The SW and LW statistics of the area ratio for the different sampling times under different sampling frequencies. The calculation moment for SW and LW is 17 October 2017 and 5 October 2017, respectively.

Times		1	2	3	4	5	6	7	8	≥9
SW	1 h	0.39%	0.32%	0.48%	0.69%	1.03%	1.45%	2.2%	3.51%	82.28%
	2 h	0.76%	1.46%	3.08%	9.27%	65.17%	12.37%	0	0	0
	4 h	3.42%	44.00%	44.22%	0	0	0	0	0	0
	6 h	33.08%	57.82%	0	0	0	0	0	0	0
LW	1 h	0.07%	0.09%	0.11%	0.29%	0.40%	0.41%	0.44	0.66%	97.52%
	2 h	0.18%	0.57%	0.83%	1.58%	19.27%	72.78%	4.76%	0	
	4 h	1.09%	11.51%	84.84%	2.42%	0	0	0	0	
	6 h	8.51%	89.38%	1.66%	0	0	0	0	0	

## Data Availability

The Jet Propulsion Lab (JPL) Planetary and Lunar Ephemerides data used in this study were downloaded from the ephemeris system at https://ssd.jpl.nasa.gov/?planet_eph_export (accessed on 30 October 2020). The datasets of CER_SYN1deg-1Hour_Terra-Aqua-MODIS_Edition4A and CER_SYN1deg-1Hour_Terra-MODIS_Edition4A were downloaded from the online CERES data ordering system at https://asdc.larc.nasa.gov/project/CERES (accessed on 30 October 2020).

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
