# Peer review of "Effect of Temporal Sampling Interval on the Irradiance for Moon-Based Wide Field-of-View Radiometer"

_sensors, 2022, doi:10.3390/s22041581_

Round 1

Reviewer 1 Report

This paper analyses Earth's outward radiative flux observed by a Moon-based Earth radiation observation. This is an interesting topic, however, there are still several defects need to be clarified by the authors.

The First is the radiant resolution. The radiant resolution herein refers to the minimum detectable energy. It is not reliable to determine the minimum detectable energy by the metrics hourly, daily, and monthly mean radiation. Actually, the minimum detectable energy must be determined based on scientific goals.It is related to another core question: how to use the measured data? What is the authors' scientific goal?  

Secondly, what is the authors' purpose in analysing temporal spatial coverage?

Thirdly, the OSR may be not a suitable parameter for Moon-based Earth radiation observations due to its significant effects of Earth's phase.

Author Response

Response to Reviewer 1 Comments

Point 1: The First is the radiant resolution. The radiant resolution herein refers to the minimum detectable energy. It is not reliable to determine the minimum detectable energy by the metrics hourly, daily, and monthly mean radiation. Actually, the minimum detectable energy must be determined based on scientific goals. It is related to another core question: how to use the measured data? What is the authors' scientific goal?  

Response 1: Thanks for your comments. Moon-based Earth observation is still in the feasibility demonstration stage and the analysis of potential applications reveal that there are several observation objectives, such as Solid earth dynamics, Earth’s environmental elements and Energy budget of Earth (Guo et al., 2018; Ye et al., 2020). The aim of Moon-based Wide Field-of-View (MWFOV) radiometer is to achieve the longer-term continuous multi-angle measurement for the Earth's outward radiative flux, which can alleviate the insufficiency of satellite platform in sampling frequency, temporal duration and spatial coverage. The scientific goal of the measurement of the Earth’s outgoing radiation for a wide field-of-view radiometer is to for analyze Energy budget of Earth, which is closely related to temporal sampling Interval. Because the sampling interval represents the time difference that the adjacent samplings have. A shorter sampling interval would result in more samples every day, but the radiometer would need a more precise design and have a higher cost. Since ERB measurement equipment is basically a heat detector, its sampling frequency is quite different, usually a few seconds (the Clouds and Earth’s Radiant Energy System (CERES)) or a few minutes (the Geostationary Earth Radiation Budget (GERB)) or even a few hours (the National Institute of Standards and Technology Advanced Radiometer (NISTAR) onboard the Deep Space Climate Observatory (DSCOVR)). The order of magnitude of entrance pupil irradiance for the MWFOV radiometer is 10-2 W∙m-2, and the magnitudes are similar as the ERBE, CERES and GERB instruments. The magnitude of irradiance measured by the NISTAR on DSCOVR is on the order of 10-3W∙m-2 and the accuracy of the detector is 1.5% or better [12], providing a meaningful reference and verification for the realization of the MWFOV radiometer. In summary, the determination of the minimum detectable energy by the metrics hourly, daily, and monthly mean radiation will facilitate the ascertainment of sampling scheme of a MWFOV under a certain manufacturing budget.

In addition, since the early 1970s, great efforts have been made to deal with these data obtained on satellite platforms at different heights, such as the Earth Radiation Budget Satellite (ERBS) and the Clouds and the Earth’s Radiant Energy System (CERES); furthermore, the corresponding complete methods for estimating the Earth's radiation budget using satellite-based data have been developed. Therefore, the corresponding Moon-based data processing method can be developed based on the processing method for the satellite-based data after combining with the characteristics of the Moon-based platform (Su et al., 2020, 2015). Besides, by analyzing the simulation time series, a very useful result for the study of the Earth energy balance can be obtained, as shown in Figure 1. The overall trend analysis of the time series reveals that the Earth's outgoing longwave radiation is increasing year by year, while the outgoing shortwave radiation is decreasing year by year. This result is consistent with the latest research results of CERES (Loeb et al., 2018), and the correlation coefficients of the results is 1, see Table 1.

In addition, we have re-edited and checked the entire manuscript in detail, and undergone extensive English revisions using the editing services recommended by the journal editorial department.

Figure 1 The overall trend of simulated original irradiance time series from March 2000 to December 2020.

Table 1 The correlation coefficients of the simulated original irradiance time series and CERES monthly outgoing radiation data from the EBAF-TOA Ed4.0 product. OLR: Outgoing longwave radiation. OSR outgoing shortwave radiation

CERES-OLR

CERES-OSR

Moon-Based-OLR

1

-1

Moon-Based-OSR

-1

1

Guo, H., Liu, G., & Ding, Y. (2018). Moon-based Earth observation: scientific concept and potential applications. International Journal of Digital Earth, 11(6), 546-557. DOI: 10.1080/17538947.2017.1356879

Ye, H., Zheng, W., Guo, H., & Liu, G. (2020). Effects of Temporal Sampling Interval on the Moon-Based Earth Observation Geometry. IEEE Journal of Selected Topics in Applied Earth Observations and Remote Sensing, 13, 4016-4029. DOI: 10.1109/JSTARS.2020.3008521

Su, W., Minnis, P., Liang, L., Duda, D.P., Khlopenkov, K., Thieman, M.M., Yu, Y., Smith, A., Lorentz, S., Feldman, D. and Valero, F.P., 2020. Determining the daytime Earth radiative flux from National Institute of Standards and Technology Advanced Radiometer (NISTAR) measurements. Atmospheric Measurement Techniques, 13(2), pp.429-443. DOI: 10.5194/amt-13-429-2020.

Su W, Corbett J, Eitzen Z, Liang L. Next-generation angular distribution models for top-of-atmosphere radiative flux calculation from CERES instruments: Methodology. Meas. Tech. 2015 Feb 5;8(2):611-32. DOI: 10.5194/amt-8-611-2015.

Loeb, N. G., D. R. Doelling, H. Wang, W. Su, C. Nguyen, J. G. Corbett, L. Liang, C. Mitrescu, F. G. Rose, and S. Kato, 2018: Clouds and the Earth’s Radiant Energy System (CERES) Energy Balanced and Filled (EBAF) Top-of-Atmosphere (TOA) Edition-4.0 Data Product. J. Climate, 31, 895-918, DOI: 10.1175/JCLI-D-17-0208.1.

Point 2: Secondly, what is the authors' purpose in analyzing temporal spatial coverage?

Response 2: Thanks for your comments. The sampling temporal sequence at a certain sampling interval would also have consequences for the measurement error, since it could affect the final temporal interpolation that derives the continuous data. Besides, the sampling frequency is closely related to the response time of the detection element. The smaller the response time, the higher the sampling frequency. In addition, the variation in the orbits of the Sun, Earth and Moon also means that the spatial coverage of the radiometer change with the different sampling temporal sequence. However, the effects of the sampling interval on the irradiance for Moon-based Wide Field-of-View (MWFOV) radiometer remain poorly understood. Therefore, the effect of temporal sampling interval on the irradiance for Moon-based wide field-of-view radiometer is analyzed in this work, and the derived results in this study could facilitate the ascertainment of sampling scheme of a MWFOV under a certain manufacturing budget and measurement error limit.

Point 3: Thirdly, the OSR may be not a suitable parameter for Moon-based Earth radiation observations due to its significant effects of Earth's phase.

Response 3: Thanks for your comments. The shortwave irradiance (OSR) ranges from about 0.00 to 94.55 mW∙m-2, while for the longwave irradiance (OLR), the range is 58.05 ~ 86.92 mW∙m-2. Since the influence of Earth's phase, Moon-based sensors cannot receive outgoing shortwave radiation (OSR) for some time. Compared with OSR, the changes in outgoing longwave radiation (OLR) are more stable and can be observed all the time. However, compared to satellite-based platforms, the variations of the Earth's phase and Moon's orbit provide a special perspective for the observation of the Earth's outgoing radiation. As the Earth moves around the Sun with its rotational axis tilted about 23.5° from the ecliptic plane and the Moon also changes with an inclination about 5.15° to the ecliptic plane, In consequence, the maximum inclination of the Moon varies from 18.35° (i.e. 23.5°−5.15°) to 28.65° (i.e. 23.5°+5.15°) (Guo et al., 2020). Besides, with the change of the Earth's phase, Moon-based detectors can sequentially capture the changing angle and radiation characteristics of the Sun's illumination area in different Earth regions (Liao et al., 2021). The above-mentioned characteristics indicate that the Moon-based Earth radiation observation can provide longer-term continuous multi-angle measurements of OSR, which will be helpful for the development and verification of angular distribution models. In addition, the Moon-based Earth observation is an important application objective for China’s future lunar exploration program, which will provide an opportunity for the acquisition of Moon-based Earth outgoing radiation observation data (Li et al., 2019). Therefore, the OSR could be regarded as a parameter of Moon-based Earth radiation observation observatory.

Guo, H., Ye, H., Liu, G., Dou, C., & Huang, J. (2020). Error analysis of exterior orientation elements on geolocation for a Moon-based Earth observation optical sensor. International Journal of Digital Earth, 13(3), 374-392.

Liao, J., Yuan, L., & Nie, C. (2021). A Simulation Method for Thermal Infrared Imagery from Moon-Based Earth Observations. IEEE Sensors Journal21(6), 7736-7747.

Li, C., Wang, C., Wei, Y. and Lin, Y., 2019. China’s present and future lunar exploration program. Science, 365(6450), pp.238-239.

Reviewer 2 Report

The manuscript examined how temporal sampling interval of a non-scanning radiometer located on the Moon surface could affect the accuracy of the observed radiation fields, based on a radiative transfer function and assumed geometrical arrangement of the Moon, the Sun, and the Earth. The idea is worthwhile, but the methodology is problematic and the results hardly give any useful information for the instrument design; besides, the manuscript was written so carelessly that there are so many wrong sentences, grammar errors, and confusing expressions. I would recommend a rejection of the manuscript for the publication on Sensors.

Specific comments:

  1. The key problem of the methodology is that the sampling interval examined are all larger than 1 hour, because of the reference CERES data which is 1-hourly. For any instrument set on the moon, it would be too coarse an interval if it can only observe 1 or several hourly. Therefore, this renders the examination of the manuscript meaningless in terms of its practical value.
  2. The figures and discussions themselves are unnecessarily tedious. As many as 22 figures are presented, but some of them (e.g., figs. 17-19, table 3) add little or no additional information that could help about the understanding of examined issue.
  3. 4 gives a flowchart of the examine process. However it is hard to read. For instance, there are two ‘yes or no’ judgement in the flowchart, but what is the question that should be judged and yield a yes or no answer?
  4. There are many broken or simply wrong sentences, illustrating that the authors are careless about the manuscript or failed to check it up before the submission. Just list a few of them:
    1. Lines 31-33: ‘the balance between the incoming solar radiation between the outgoing radiation of the reflection of solar shortwave (SW) radiation and the emission of longwave infrared (LW) radiation.’
    2. Lines 33-34: ‘Due to the ERB quantifies how the earth gains energy from the Sun and loses energy to space; it is…’
    3. Lines 96-97: ‘Then the irradiance will be converted a digital signal and the digital signal will be will be processed and saved by the Electrical system’.
    4. Lines 102-103: ‘model simulations based the geometric conditions [18, 31] and potential applications of Moon-based Earth observations [32, 33].’
    5. Lines 121-123: ‘In this work, based on the based on radiation transfer model, the effect of temporal sampling interval’

  1. There are also countless grammar and spelling errors that I would not mention one by one here. The authors should at least correct those errors before the submission.

Author Response

Response to Reviewer 2 Comments

Point 1: The key problem of the methodology is that the sampling interval examined are all larger than 1 hour, because of the reference CERES data which is 1-hourly. For any instrument set on the moon, it would be too coarse an interval if it can only observe 1 or several hourly. Therefore, this renders the examination of the manuscript meaningless in terms of its practical value. 

Response 1: Thanks for the comments. The aim of Moon-based Wide Field-of-View (MWFOV) radiometer is to achieve a longer-term continuous multi-angle measurement for the Earth's outward radiative flux. The scientific goal of the measurement of the Earth’s outgoing radiation for a wide field-of-view radiometer is to for analyze Energy budget of Earth, which is closely related to temporal sampling Interval. Because the sampling interval represents the time difference of the adjacent samplings have. Besides, the sampling frequency is closely related to the response time of detection element. The smaller the response time, the higher the sampling frequency. A shorter sampling interval would result in more samples each day, but the radiometer would need a more precise design and have a higher cost. The sampling temporal sequence at a certain sampling interval would also have consequences for the measurement error, since it could affect the final temporal interpolation that derives the continuous data.

The outgoing radiation is the key part of research into the Earth Radiation Budget (ERB) at the Earth’s top of the atmosphere. Since ERB measurement equipments are basically the heat detector, its sampling frequency is quite different (see Table 1), usually a few seconds (the Clouds and Earth’s Radiant Energy System (CERES)) or a few minutes (the Geostationary Earth Radiation Budget (GERB)) or even a few hours (the National Institute of Standards and Technology Advanced Radiometer (NISTAR) onboard the Deep Space Climate Observatory (DSCOVR)). Current LEO ERB systems could not record the rapid variability of the OLR and OSR for the Earth atmosphere system due to the limited temporal sampling coverage. However, the temporal sampling coverage of a MWFOV is far higher than the LEO counterparts. The sampling period of a MERO is approximately 12 per day for a fixed location on Earth if the sampling interval is set to 60 min. Similarly, it is 48 temporal samples per day if the sampling interval is set to 15 min (the GERB sampling interval). These abundant temporal samples would substantially enhance the quality of the diurnal data produced by the ERB fitting method and could help to reveal the small-temporal-scale variations of the top of atmosphere (TOA) outgoing longwave radiation (OLR) and outgoing shortwave radiation (OSR).

Finally, the CERES SYN1deg-1Hour Ed4A products is designed to provide the highest temporal resolution TOA flux dataset by incorporating hourly geostationary-Earth-orbit (GEO) imager data and by taking advantage of the additional GEO imager channels to improve the cloud property retrievals, computed surface fluxes, and GEO-derived TOA fluxes. To quantify the effect of temporal sampling interval on the irradiance for Moon-based Wide Field-of-View radiometer, here we use the CERES SYN ED4.1A dataset to construct the true OSR and OLR fluxes based on the radiation transfer model. Therefore, the derived results in this study could facilitate the ascertainment of a sampling scheme of a MWFOV under a certain manufacturing budget and measurement error limit.

  Table 1 The parameters of satellite-based platform for observing the outgoing radiation from the Earth’s top of atmosphere. ERBE: the Earth Radiation Budget Experiment

Name

Orbit height

Temporal resolution

CERES

700-712km

<1s / Global daily

ERBE

800-950km

<1s / Global daily

GERB

3.6×104km

15min

DSCOVR

1.5×106km

1-1.6 h(EPIC)

10min/30min/4h(NISTAR)

Guo, H., Liu, G., & Ding, Y. (2018). Moon-based Earth observation: scientific concept and potential applications. International Journal of Digital Earth, 11(6), 546-557. DOI: 10.1080/17538947.2017.1356879

Ye, H., Zheng, W., Guo, H., & Liu, G. (2020). Effects of Temporal Sampling Interval on the Moon-Based Earth Observation Geometry. IEEE Journal of Selected Topics in Applied Earth Observations and Remote Sensing, 13, 4016-4029. DOI: 10.1109/JSTARS.2020.3008521

Point 2: The figures and discussions themselves are unnecessarily tedious. As many as 22 figures are presented, but some of them (e.g., figs. 17-19, table 3) add little or no additional information that could help about the understanding of examined issue.

Response 2: Thank you for your reviewing and commenting. We have condensed and reorganized this manuscript, mainly including introduction, results, discussion and conclusion. In section of results, the discussions about figures and tables are further clarified, and the table 3 is deleted.

Point 3: Figure 4 gives a flowchart of the examine process. However, it is hard to read. For instance, there are two ‘yes or no’ judgement in the flowchart, but what is the question that should be judged and yield a yes or no answer?

Response 3: Thank you for reviewing our manuscript. We have made corrections about the flowchart for the calculation of irradiance in Figure 4 and a detailed description is given in section 2.4. (L304-318)

L304-318: “The flowchart for the calculation of the simulated original irradiance time series of the MWFOV radiometer is shown in Figure 4. The first step is to set the time range, time step and lunar surface position. The second step is to discretize the Earth’s TOA at a resolution of 1° × 1° and acquire the coordinate vector in the same coordinate and time system based on the Moon-based Earth observation geometry. The third step determines the MWFOV-radiometer-viewed node area at the Earth’s TOA. The fourth step solves the radiation transfer function to obtain the MWFOV radiometer’s entrance pupil irradiance at a certain moment. The fifth step judges whether the calculation is completed at all time points. If yes, the irradiance is output; if no, then the next time calculation is entered−that is, the model returns to the second step. Finally, the simulated irradiance time series is output and the calculation process is ended. Due to the maximum difference between different lunar locations being 9×10−4 W∙m−2, which is too small to remarkably improve the observation performance of the platform, the lunar surface site 0°E0°N is selected as the position of the MWFOV radiometer for the study of the effect of the temporal sampling interval on irradiance.”

Figure 4. Flowchart for the calculation of irradiance. ADMs: angular distribution models

Point4 &5: There are many broken or simply wrong sentences, illustrating that the authors are careless about the manuscript or failed to check it up before the submission. There are also countless grammar and spelling errors that I would not mention one by one here. The authors should at least correct those errors before the submission.

Response 4&5: Thank you for reviewing our manuscript. We have re-edited and checked the entire manuscript in detail, and undergone extensive English revisions using the editing services recommended by the journal editorial department.

Round 2

Reviewer 1 Report

Having read authors' response of the comments. The authors still have not clarified what the scientific goal is. How to use the MWFOV radiometer data to realize the Energy budget of the Earth? When observing from the Moon, half of the Earth is always invisible. How to break through this limitation?  The authors should address this issue and then do more detailed research.

Author Response

Point 1: Having read authors' response of the comments. The authors still have not clarified what the scientific goal is. How to use the MWFOV radiometer data to realize the Energy budget of the Earth? When observing from the Moon, half of the Earth is always invisible. How to break through this limitation?  The authors should address this issue and then do more detailed research. 

Response 1: Thanks for your comments. The scientific goal of the measurement of Earth’s outgoing radiation by a Moon-based wide field-of-view radiometer is to obtain large-scale, continuously changing observation angles and long-term time series observations data, which will improve the quality of data on the outgoing radiation of the Earth (Guo et al., 2018). Then, the data can be used to analyze the Earth Radiation Budget (ERB) at the top of the atmosphere and the ERB can be measured by the balance of the Earth gains energy from the Sun (F0) against energy lost to space through the outgoing shortwave radiation (OSR) and the outgoing longwave radiation (OLR). According to the conservation of energy, we can get the equation (1) (Liang, S., 2018):

(1)

where Fnet is the net radiation on the Earth’s TOA between incoming and outgoing radiative fluxes.

   Limited in its own synchronous rotation, the nearside of the Moon always faces Earth. Besides, since the rotation cycle of the Earth is 24 hours, therefore, the nearside of the moon can sample the whole Earth in one day, which will be helpful to overcome the shortcomings of half of the Earth being always invisible from the Moon. Figure 1 shows that the longwave (5-200 μm) coverage images at the sampling interval of 1hour for the Moon-based Wide Field-of-View (MWFOV) radiometer on October 5, 2017.

Figure 1. The outgoing longwave (5-200 μm) radiation coverage image times at the sampling interval of 1hour for the MWFOV radiometer on October 5, 2017.

The result in Figure 1 reveals that most of regions on the Earth’s top of atmosphere (TOA) can be observed 12 times in one day if the sampling interval is set to 60 min and almost the entire earth can be observed in one day. Similarly, it is 48 temporal samples per day if the sampling interval is set to 15 min. The sampling interval represents the time difference that the adjacent samplings have. A shorter sampling interval would result in more samples every day, but the radiometer would need a more precise design and have a higher cost. The sampling frequency is also closely related to the response time of the detection element. The smaller the response time, the higher the sampling frequency. Since ERB measurement equipment is basically a heat detector, its sampling frequency is quite different (See table 1), usually a few seconds (the Clouds and Earth’s Radiant Energy System (CERES)) or a few minutes (the Geostationary Earth Radiation Budget (GERB)) or even a few hours (the National Institute of Standards and Technology Advanced Radiometer (NISTAR) onboard the Deep Space Climate Observatory (DSCOVR)). The MWFOV radiometer for acquiring the Earth’s outgoing radiation data is closely related to temporal sampling Interval. However, the effects of the sampling interval on the irradiance for the Moon-based Wide Field-of-View (MWFOV) Radiometer remain poorly understood. Therefore, the effect of the temporal sampling interval on the irradiance for the Moon-based Wide Field-of-View radiometer is analyzed in this work, and the derived results could facilitate the ascertainment of a sampling scheme for the MWFOV radiometer under a certain manufacturing cost.

Table 1: The parameters of satellite-based platform for observing the outgoing radiation from the Earth’s top of atmosphere. ERBE: the Earth Radiation Budget Experiment

Name

Orbit height

Temporal resolution

CERES

700-712km

<1s / Global daily

ERBE

800-950km

<1s / Global daily

GERB

3.6×104km

15min

DSCOVR

1.5×106km

1-1.6 h(EPIC)

10min/30min/4h(NISTAR)

In addition, in order to clarify the scientific goal of this manuscript, we have revised the introduction and discussion.

Guo, H., Liu, G., & Ding, Y. (2018). Moon-based Earth observation: scientific concept and potential applications. International Journal of Digital Earth, 11(6), 546-557. DOI: 10.1080/17538947.2017.1356879

Liang, S. (2018). Volume 5 overview: recent progress in remote sensing of earth's energy budget. Comprehensive Remote Sensing, 5, 1-31.

Reviewer 2 Report

Basically, the lauguage of the manuscript has been significantly improved in this revised version, and the main points are much more clearly demonstrated now. But some improvements, both in language and descriptions, should be made before it is suitable for publication.

Descriptions:

  1. Lines 188-189: it’s confusing to name the two longwave spectral ranges as “the infrared thermal emission” and “the outgoing longwave radiaiton”, since these two terms are conventionally referring to the same thing. What the authors actually mean here may be the whole longwave spectrum (OLR) and the window region (OLR-NW, or 8-12 μm), as shown in the following discussions. It is suggested that the three spectral ranges of the irradiances analized should be more specifically given as numbers, rather than only termed.
  2. Following the question above, another problem is that the spectral range of OLR is wrongly given. As shown in Fig. 5 and other figures, the OLR range is 0.2-200 μm, this is definitely incorrect. It should be about 4-200 μm, while the range of 0.2-4 μm is OSR. If the range is wrongly selected, then all the resutls in the manuscript are not correct.
  3. Lines 341-342: It’s better to point out the time samples listed are for a sampling frequency of 2-hourly and 3-hourly, respectively.
  4. Line 352: It seems “described by the uncertainty in subsampling” should be “described as the uncertainty in subsampling”. The variation is a quantity directly derived from computation, while uncertainty is what the variation represents.
  5. Line 368: It's confusing which two quantities are compared here. It's unclear what is the 'calculation range of the mean' (the range of the mean values). The authors may mean: if the sampling interval is shorter than the period considered, this will allow for multiple sampling.
  6. Lines 415-416: The legends on figure 6(a) seem to have assigned wrong colors to OSR and OLR. The OSR (red line) has no peak around Oct 18th as described in these two lines.
  7. Figure 10: The OLR spectral range (2-200 μm) is inconsistent with those in other places of the manuscript, although all these values are wrong.
  8. Lines 544-546: The 3-sigma principle is introduced here, but only actually used in the lines starting from line 608. It would be more coherent to move it to around line 608.
  9. Figure 13: It's not mentioned how the normal distribution curves are derived. Are they ploted according to normal distribution functions with the given mu and sigma values in the figures?
  10. Figures 17-19: The mu values in all the three figures are 0, meaning that no matter what sampling intervals are usesd, the mean irradiances obtained are exactly the same as the true original ones. This doesn't make sense.

Language and typo:

  1. Line 155: “the all of results” should be “all of the results”.
  2. Lines 178-179: “the one of major sources” should be “one of the major sources”.
  3. Equation 4: the has a missing subscript i.
  4. Line 277: “These datasets” should be “This dataset”.
  5. Line 304: “equation (5) and (6)” should be “equations (8) and (9)”.
  6. Line 334: “the the simulated” should be “the simulated”.
  7. Figure 9(a): The spectral range of OSR should be 0.2–5, not 2–5 μm.
  8. Figure 9(e): The red square is located out of the line.
  9. Figure 11: The range of OLR-NW is 8-12 μm, rather than 2-200 μm.
  10. Line 521: “on the end of each month” should be “at the end of each month”.
  11. Figure 13: The units of the x-axis should be given, to avoid the confusion that it stands for the normalized STD, which it’s not.
  12. Line 670 and all the following text: “Table 4” should be “Table 3”.
  13. Line 801: “state of art” should be “state of the art”.

Author Response

Point 1: Lines 188-189: it’s confusing to name the two longwave spectral ranges as “the infrared thermal emission” and “the outgoing longwave radiation”, since these two terms are conventionally referring to the same thing. What the authors actually mean here may be the whole longwave spectrum (OLR) and the window region (OLR-NW, or 8-12 μm), as shown in the following discussions. It is suggested that the three spectral ranges of the irradiances analyzed should be more specifically given as numbers, rather than only termed. 

Response 1: Thank you for your comment and suggestion. The infrared thermal emission and the outgoing longwave radiation are the same. We have revised the expressions and the related contents located lines 178−180 in section 2.1.

Point 2: Following the question above, another problem is that the spectral range of OLR is wrongly given. As shown in Fig. 5 and other figures, the OLR range is 0.2-200 μm, this is definitely incorrect. It should be about 4-200 μm, while the range of 0.2-4 μm is OSR. If the range is wrongly selected, then all the results in the manuscript are not correct.

Response 2: Thank you for your comment. The spectral range of OLR and OSR are 5-200 μm and 0.2-5 μm, respectively, and the results in the manuscript is based on the spectral ranges. Fig.5 and relevant expressions in the manuscript have been revised in line 352−356.

Point 3: Lines 341-342: It’s better to point out the time samples listed are for a sampling frequency of 2-hourly and 3-hourly, respectively.

Response 3: Thank you for reviewing our manuscript. We have revised the statement and added an explanation in lines 336−337 of section 2.5.1.

Point 4: Line 352: It seems “described by the uncertainty in subsampling” should be “described as the uncertainty in subsampling”. The variation is a quantity directly derived from computation, while uncertainty is what the variation represents.

Response 4: Thank you for reviewing our manuscript. We have revised the statement according to your suggestion in line 345.

Point 5: Line 368: It's confusing which two quantities are compared here. It's unclear what is the 'calculation range of the mean' (the range of the mean values). The authors may mean: if the sampling interval is shorter than the period considered, this will allow for multiple sampling.

Response 5: Thank you for reviewing our manuscript. Indeed, it is confusing which two quantities are compared here for readers and the related expressions are deleted. In fact, we have explain the means in lines 363-365.

Point 6: Lines 415-416: The legends on figure 6(a) seem to have assigned wrong colors to OSR and OLR. The OSR (red line) has no peak around Oct 18th as described in these two lines.

Response 6: Thank you for your comment. We have modified legends in the Figure 6.

Point 7: Figure 10: The OLR spectral range (2-200 μm) is inconsistent with those in other places of the manuscript, although all these values are wrong.

Response 7: Thank you for reviewing our manuscript. We have modified the Figure 10, and checked all figures and spectral ranges in the manuscript.

Point 8: Lines 544-546: The 3-sigma principle is introduced here, but only actually used in the lines starting from line 608. It would be more coherent to move it to around line 608.

Response 8: Thank you for reviewing our manuscript. The 3-sigma principle are used in both places (Lines 535−545 and Lines 603−611). We added a description at around line 603 in order to express more clearly.

Point 9: Figure 13: It's not mentioned how the normal distribution curves are derived. Are they plotted according to normal distribution functions with the given mu and sigma values in the figures?

Response 9: Thank you for reviewing our manuscript. The normal distribution curves in Figure 13 are plotted according to normal distribution functions with the given mean, mu, and standard deviation, sigma. Besides, the description about the curves is added in title of Figure 13.

Point 10: Figures 17-19: The mu values in all the three figures are 0, meaning that no matter what sampling intervals are used, the mean irradiances obtained are exactly the same as the true original ones. This doesn't make sense.

Response 10: Thank you for reviewing our manuscript. The mu values in Figures 17-19 is the mean of absolute error for the simulated subsampled time series, so the mu values equal to 0 in all the three figures. Besides, they are deleted in the manuscript.

Point 11: Language and typo.

Response 11: Thank you for reviewing our manuscript, and we have re-edited and checked the entire manuscript in detail.

Round 3

Reviewer 1 Report

Though I accept this manuscript in present form, the authors still have not stated the scientific goal of the Energy budget by using the data from the Moon. Hope to do more efforts on it in the follow up research.

Reviewer 2 Report

All my concerns have been addressed properly, I would suggest its publication on Sensors after doing the following:

The authors should check up the correctness of the equation numbers in line 296: 'where M (θ0, i) and M (γi, ti) in the equation (5) and (6) ...' (should be 8 and 9?).